# VLA-in-the-Loop: Online Policy Correction with World Models for Robust Robotic Grasping

## Abstract

Large-scale Vision-Language-Action (VLA) models excel at mapping natural language instructions to robotic action. However, they typically treat actions as terminal outputs with imitation learning often leads to execution bias, lacking mechanisms for dynamic supervision or online error correction. Meanwhile, World models (WM) have shown promise for predictive reasoning, but prior approaches typically require continuous frame-by-frame rollout of long sequences, resulting in high computational cost and limited flexibility. In this work, we propose VLA-in-the-Loop, a novel framework that introduces an online intervention mechanism to correct the base VLA policy. Our core innovation lies in the use of a lightweight, composite World Model, not for continuous state prediction, but as an on-demand, event-triggered "corrector." When the VLA proposes a high-stakes action (e.g., closing the gripper), at this critical juncture, our composite WM first employs its discriminative component to evaluate the action's feasibility. Should the proposed action be deemed unviable, a generative model synthesizes a short video of a successful future trajectory from the current state. Robot will be guided to the correct position using actions decoded by inverse dynamics mode(IDM) and execute a corrected, more robust action. This plug-in architecture is not only computationally efficient but also enhances data utilization by learning from potential failures, thereby significantly improving the robustness of VLA models against online disturbances. We validate our framework across multiple robotic grasping tasks on both simulation and real-world systems, demonstrating the effectiveness of using world models not only for prediction, but as active agents for real-time correction in VLA-based robotic systems.

## 1 Introduction

Vision-Language-Action (VLA) models  (Brohan et al., 2022; Zitkovich et al., 2023; O'Neill et al., 2024; Team et al., 2024; Kim et al., 2024; Qu et al., 2025; Zheng et al., 2024; Li et al., 2024b;a) have emerged as a powerful paradigm for robotic manipulation, enabling agents to interpret natural language instructions and generate corresponding actions from visual input. Despite their success, these systems rely heavily on imitation learning on *positive grasp data*, treating action generation as a terminal step and lacking mechanisms for dynamic supervision or online correction. This often results in execution bias, particularly in dynamic or ambiguous environments where small perturbations can lead to critical failures.

Parallel to the development of VLAs, world models (WMs) (Ha & Schmidhuber, 2018; Agarwal et al., 2025; Wu et al., 2025) have gained significant attention for their ability to model environmental dynamics and support predictive reasoning. By simulating future states, these models enable agents to anticipate consequences and plan accordingly. However, prior methods typically rely on continuous, frame-by-frame rollouts to generate future states, a computationally expensive process impractical for real-time control.

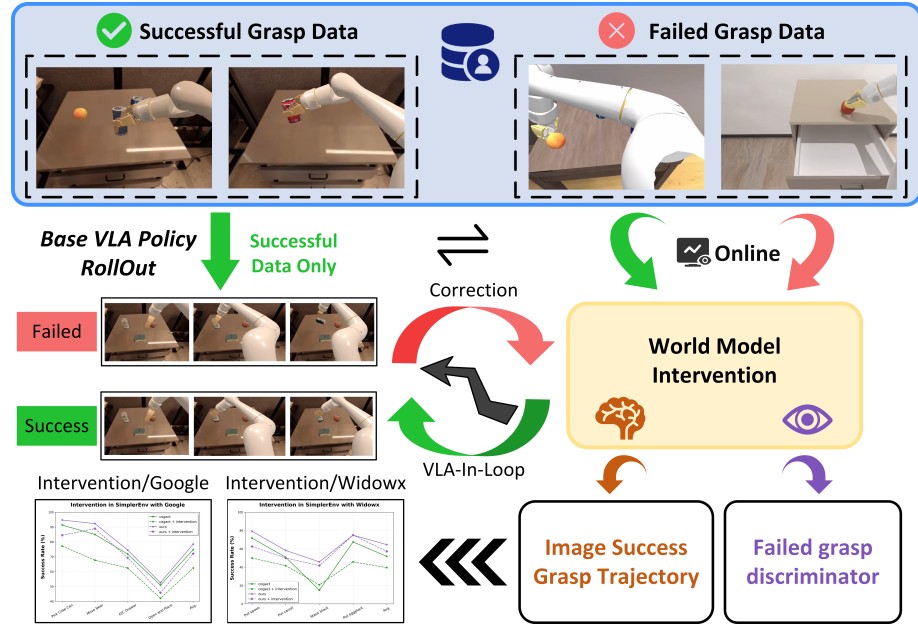

Figure 1: An overview of our core idea. Standard VLAs (left) suffer from compounding errors, leading to task failure, which is trained only on successfully collected data. Our VLA-in-the-Loop framework (right) intervenes at critical moments, evaluating the proposed action and leveraging a world model to imagine a successful outcome by incorporating both successful and failed collected data, which then guides a corrected action to ensure task success.

Moreover, prior models are often used passively as an auxiliary supervision signal during training, rather than actively guiding decision-making during execution.

To address the limitations of both paradigms, we ask: *what if an agent could pause at critical moments, assess the feasibility of its own plan, and imagine a trajectory to success before committing to a potentially flawed action?* In this work, we propose a framework for Online Policy Rectification called VLA-in-the-Loop, which synergistically incorporates the generalized capabilities of VLAs with the predictive power of a lightweight, event-driven world model. Our approach is designed to address two key limitations in current robotic manipulation pipelines: (1) the inability to robustly detect and respond to grasp failures in real time, and (2) the underutilization of world models in guiding corrective actions. Specifically, our method begins by using a primary VLA to assess whether the current visual frame constitutes a keyframe suitable for grasping. At this juncture, our framework first leverages a discriminative module to evaluate the risk of failure for the proposed action, which is trained on the data composed of **high-value failure cases**. If a potential failure is detected, a generative module "imagines" a short video of a successful grasp from the current state. This imagined success trajectory is then looped back to the VLA, serving as a clear visual guide to generate a corrected, more precise action.

This "in-the-loop" architecture is designed for synergy and efficiency. By activating the WM only when necessary, it avoids the computational overhead of continuous prediction. Crucially, by learning from potential failures, it enhances data efficiency and robustness. In summary, our contributions are as follows:

- We propose a novel and efficient integration of Vision-Language-Action (VLA) models with world models (WM), where the WM operates as a lightweight, event-driven plug-in that is selectively activated only when a grasp action is predicted. This design enables real-time correction without incurring the cost of continuous rollouts.

- Our framework is the first work to apply world models for grasp correction, leveraging their discriminative ability to assess graspability and generative capacity to predict a single future graspable state—thus avoiding the continuous resource-intensive generation used in prior methods.

- We introduce a failure-driven training strategy for the world model, utilizing both successful and failed executions to learn robust decision boundaries. This improves data efficiency and generalization, leading to state-of-the-art performance across multiple robotic grasping benchmarks with strong robustness to online intervention and environmental disturbances.

## 2 RELATED WORKS

**Vision-Language-Action (VLA) Models**   The integration of vision and language into robotic control has led to the development of Vision-Language-Action (VLA) models, which enable robots to interpret multimodal inputs and generate context-aware actions. Early pioneers like VIMA (Jiang et al., 2022) demonstrated the potential of a single, massive network to handle diverse tasks. This was followed by more specialized robotics models like RT-1 (Brohan et al., 2022), and RT-2 (Zitkovich et al., 2023), which showed that co-fining vision-language models on robotics data could transfer abstract web-scale knowledge to physical control, enabling impressive zero-shot generalization. More recent efforts, such as CogVLM (Wang et al., 2024) and CogACT (Li et al., 2024a), have integrated more powerful vision encoders, further enhancing the models' perceptual understanding. However, these models typically rely on imitation learning and treat action generation as a terminal output, which limits their adaptability in dynamic environments, the problem of online error accumulation in unseen situations remains a significant open challenge. Our work directly addresses this limitation not by changing the VLA architecture itself, but by augmenting it with an external, intelligent correction loop.

**World Models in Robotics**   World Models, which learn a predictive model of an environment, enabling agents to simulate future states and plan accordingly from current observations and actions. Seminal works like PlaNet (Hafner et al., 2019) and Dreamer series (Hafner et al., 2020) have shown that agents can learn complex behaviors by "dreaming" or planning within a learned latent space. WorldVLA (Cen et al., 2025) attempted to unify VLA and world models into a single framework, highlighting the mutual benefits of joint training. However, these models typically require continuous, step-by-step prediction of future states (often as images or latent vectors) for offline planning, serving merely as an auxiliary head to provide an extra supervision signal during training, rather than actively guiding the online correction of a powerful learning policy, lacking explicit mechanisms for failure recovery. Our key distinction is the repurposing of WM components (generation and discrimination) for on-demand, event-triggered intervention, avoiding the overhead of constant future simulation.

**Online Policy Correction and Intervention.**   The concept of online correction is crucial for deploying robots in dynamic environments. Traditional grasping pipelines (Kahn et al., 2017; Ebert et al., 2018) often rely on static policies or human intervention when failures occur, limiting their adaptability in dynamic or ambiguous environments. A prominent direction leverages the reasoning capabilities of Large Vision-Language(LVLMs) (Shi et al., 2024; Shinn et al., 2023; Luo et al., 2025) to enable robots to reflect on failed grasp attempts and adjust strategies accordingly. Other approaches, such as Phoenix (Xia et al., 2025) proposes a motion-based self-reflection framework that translates high-level semantic feedback into fine-grained action corrections, enabling precise recovery from manipulation errors. However, these methods either require human oversight or struggle to generalize to the vast number of potential failure modes. In contrast, our framework introduces a proactive, perceptually-grounded self-correction loop that leverages the predictive power of a world model. By combining discriminative evaluation and generative prediction, our system can assess graspability and simulate future graspable states in a lightweight and targeted manner.

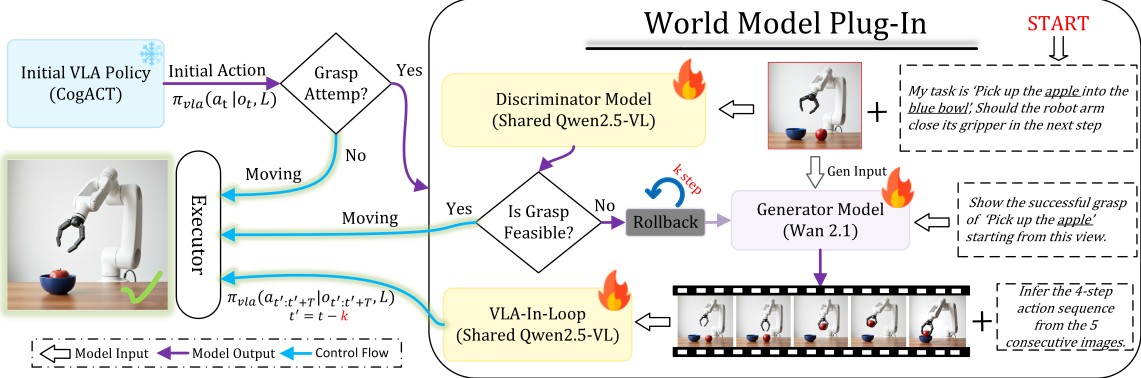

Figure 2: The architecture of our ***VLA-in-Loop*** framework. The process follows a 'Propose-Evaluate-Imagine-Correct' loop. When a keyframe (grasp initiation) is detected, a Discriminator evaluates the proposed action. If failure is predicted, a Generator generates a video of successful grasp. This video is then fed back to the VLA, which outputs a corrected, final action. Note the weight-sharing between the Discriminator and the corrective VLA module for efficiency.

## 3 METHODOLOGY

Our proposed method, ***VLA-in-the-Loop***, repurposes the concept of a World Model (WM) as a novel on-line intervention mechanism to correct a base VLA policy at critical decision points. Unlike in traditional applications where WMs continuously predict future states, our approach utilizes a lightweight, composite WM as an on-demand "corrector" that combines discriminative and generative capabilities. This section first formulates the problem of online correction, then details our VLA-in-the-Loop framework, and finally elaborates on the specific design of our composite world model components.

**Problem Formulation**    We formulate the robotic manipulation task as a Partially Observable Markov Decision Process (POMDP). Let the observation space be $\mathcal{O}$, at each timestep $t$, the agent receives a visual observation $o_t = (I_t, s_t) \in \mathcal{O}$ combines an RGB image $I_t \in \mathcal{I}$ and proprioceptive state $s_t \in \mathcal{S}$ (e.g., end-effector pose, gripper state). A pre-trained VLA provides a policy $\pi_{\text{VLA}}(a_t|o_t, L)$, trained via behavioral cloning on an expert dataset $\mathcal{D}_{\text{expert}} = \{(o_i, a_i)\}_{i=1}^N$, produces a low-level action $a_t = (\Delta xyz_t, \Delta rpy_t, g_t)$, where $(\Delta xyz_t, \Delta rpy_t)$ denotes Cartesian and orientation increments, and $g_t$ is the gripper command (open/close) which is conditioned on the observation and a language instruction $L$. The training objective minimizes the deviation from expert actions:

$$\pi_{\text{VLA}} = \arg\min_{\pi} \sum_{(o,a) \in \mathcal{D}_{\text{expert}}} \mathcal{L}_{\text{imitation}}(\pi(o, L), a)$$

The central challenge during deployment is the covariate shift, where the online observation distribution $P_{\text{online}}(o)$ deviates from the training distribution $P_{\text{expert}}(o)$. This causes the performance of $\pi_{\text{VLA}}$ to degrade, leading to an expected return $\mathbb{E}_{\tau \sim \pi_{\text{VLA}}}[\sum_t R(s_t, a_t)]$ that falls below an acceptable threshold.

To solve this, we introduce an online correction function $\mathcal{C}$, which modulates the VLA's proposal $a_t = \pi_{\text{VLA}}(o_t, I)$ to produce a final action $a_t' = \mathcal{C}(o_t, a_t)$. We instantiate our World Model as $\mathcal{C}$, which intelligently intervenes to guide the policy back toward a successful trajectory and maximizes the expected return of the resultant policy $\pi'$ while minimizing computational overhead by invoking $\mathcal{C}$ sparingly, denoted as: $\max_{\mathcal{C}} \mathbb{E}_{\tau \sim \pi'}[\sum_{t=0}^{T} \mathcal{R}(s_t, a_t')]$

### 3.1 BASE POLICY: KEYFRAME DETECTION VIA PRIMARY VLA

The first step in our framework is to leverage a pre-trained VLA model, such as CogACT, as the base policy to generate an initial expert's action, $a_t = (\Delta xyz_t, \Delta rpy_t, g_t)$. For manipulation tasks like grasping, the most decisive moment is the initiation of the grasp itself. Therefore, we define the keyframe trigger as the timestep t where the base VLA policy proposes an action with gripper state transitioning to "close". The supervisor takes control and first evaluates the proposed action's likelihood of success using a discriminative module, $D_t$, at this keyframe. For all other proposed actions, such as moving the arm or opening the gripper, the supervisor remains inactive, and the VLA's predicted action is executed directly without intervention.

### 3.2 INTERVENER: A COMPOSITE WORLD MODEL

Our WM is not a monolithic dynamics predictor but a composite system of two specialized components: a VLM-based discriminator for assessing feasibility and a video-generative model for imagining corrections.

**The Discriminator: Is This Grasp Feasible?**    The discriminative module $D_t$ is realized using a powerful Vision-Language Model (VLM), specifically a fine-tuned Qwen-VL 2.5 (Bai et al., 2025). We formulate the failure prediction task as a Visual Question Answering (VQA) problem. Instead of training a specialized binary classifier, which is data-intensive. The module receives the current observation $o_t$ and a structured text prompt: "*Query: My task is 'Stack Green Block on Yellow Block', Should the robot arm close its gripper in the next step?*" The model, denoted as $\mathcal{M}_{\text{disc}}$, is trained on a curated dataset of both successful and failed grasp attempts, learning to output $D_t \in \{$Suitable, Unsuitabel$\}$. This process can be formally expressed as maximizing the probability of the output text given the visual and text inputs:

$$D_t = \arg\max P(d|o_t, c_{\text{disc}}; \theta_{\mathcal{M}_{\text{disc}}}), d \in \{\text{suitable, unsuitable}\} \tag{1}$$

This VQA formulation allows the VLM to leverage its rich, pre-trained understanding of object properties, spatial relationships, and physical affordances to make an informed judgment about the feasibility of the proposed grasp.

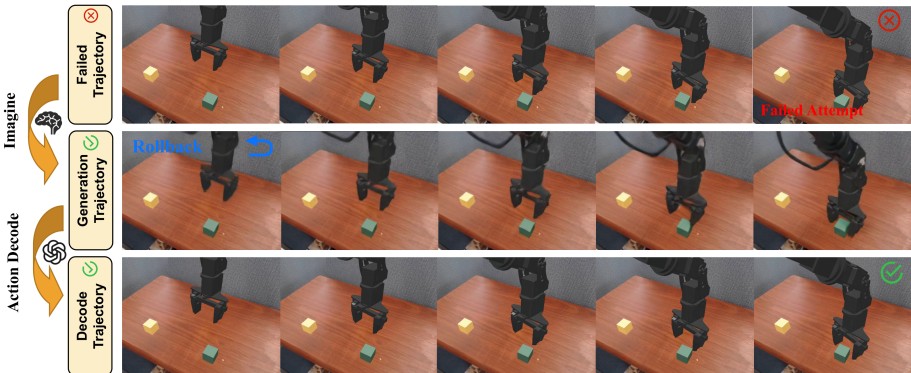

Figure 3: **The core correction mechanism of our framework**. The top row shows a failed trajectory from the base VLA policy. Upon predicting failure, our system generates a successful trajectory (middle row). This imagined plan is then decoded and executed, leading to task success (bottom row).

**The Generator**    When the discriminative component signals a potential failure ($d_t = $ unsuitable), as shown in top row in Figure 3, the system performs a state rollback of k steps to retrieve a more stable, recent observation, $o_{t'}$, where $t' = t - k$, and then activate the *Generative Model*, $\mathcal{M}_{\text{gen}}$, such as a image-to-video

diffusion model (e.g., WAN2.1 Wan (2025)). The generator receives the observation $o_{t'}$ and a high-level goal $c_{gen}$, e.g., "Show the successful grasp of 'Pick up the apple' starting from this view." It then synthesizes a short, physically plausible video sequence, $V_{succ} = (o_{t'}, o_{t'+1}, \ldots, o_{t'+T})$ that depicts a successful trajectory. This generation process is a conditional sampling from the model's learned distribution:

$$V_{succ} \sim P(V|o_{t':t'+T}, c_{gen}; \theta_{\mathcal{M}_{gen}}) \tag{2}$$

### 3.3 VLA-IN-THE-LOOP FOR ACTION REFINEMENT

The final and most crucial step of our framework is closing the corrective loop by feeding the generated knowledge back into the policy. We take the imagined successful video $V_{succ}$, present in middle row in Figure 3, serves as a new, potent visual context for updating the new VLA policy $\pi'_{VLA}(a'_t|o'_t \in v_{succ}, L)$. To maximize resource efficiency, we architect the discriminative module ($\mathcal{M}_{disc}$), which also serves as our behavior parser, acting as the final VLA by sharing its weights. We enable this dual-role capability through a multi-task Question-Answering (QA) training scheme, where the model is trained on different QA formats corresponding to its roles as an evaluator and an actor, details can be found in Figure 3 in Appendix A.2.3. This efficient "VLA-in-the-loop" approach, where the same model is used to both judge and act, establishes a tight coupling between evaluation and action, resulting in a refined action with a significantly higher likelihood of success, denote as bottom row in Figure 3.

### 3.4 TRAINING OBJECTIVES

Our framework is optimized holistically through a composite training objective designed to supervise each part of the rectification process. For the base VLA policy, we do not need extra supervision loss in training. $\mathcal{L}_{act\_corrected}$, ensures the refined action accurately follows the corrective path after conditioning on the imagined future. Concurrently, the world model components are trained to support this process. The discriminative VLM is supervised with a binary cross-entropy loss, $\mathcal{L}_{disc}$, for the keyframe classification task, while the generative video model is trained with a reconstruction loss, $\mathcal{L}_{gen}$, to ensure the fidelity of the imagined successful trajectories.

## 4 EXPERIMENTS

### 4.1 EXPERIMENTAL SETUP

**Experimental Platforms** We evaluated our framework across both real-world and simulated environments. For real-world validation, we used five challenging long-horizon tasks on two distinct dual-arm platforms: the Xiaomi Robot, a wheeled system with two 7-DoF arms, and ALOHA, featuring two 6-DoF arms with parallel grippers. All real-world experiments were performed on a single NVIDIA RTX 3090 GPU, with 32 trials conducted per task. For simulation, we conducted experiments on two platforms—SIMPLER (Li et al., 2024c) and LIBERO Liu et al. (2023)—utilizing three types of robotic arms: WidowX, Google Robot, and Franka.

**Initial VLA policy** The initial VLA policy adopts the pre-trained CogACT Li et al. (2024a) model. Crucially, we do not fine-tune this initial policy. This is a deliberate design choice, as fine-tuning actors typically requires a large amount of successful demonstration data, which is difficult and expensive to collect, especially for real robots. Our method instead prioritizes the use of failure data to train the world model for correction. The initial policy predicts a 15-step future action sequence, which serves as the fundamental proposal for the subsequent correction process.

**World Model Training** Our world model consists of a discriminator and a generator. For the discriminator, we created a targeted dataset by manually and semi-automatically labeling critical keyframes from

demonstrations as either 'suitable' or 'unsuitable' for grasping. This dataset comprises 100k frames from a subset of BridgeV2 (Walke et al., 2023) and 2k frames collected from our real-world setups, with a balanced 1:1 ratio of positive and negative samples. An 'unsuitable' state is defined by trajectories where the gripper state indicates no grasp is intended or possible. if unsuitable state detected, the robot arm will roll back $m$ step, default with 20, to get new observation. For the generation model, we used 33k video clips from the BridgeV2 dataset. For real-world fine-tuning, we collected 200 successful grasping videos (100 each from the Xiaomi Robot and ALOHA), with each video consisting of 5-20 frames.

**VLA-in-Loop policy finetuning** We constructed a custom VQA dataset to fine-tune the unified VLA-in-Loop model with successful grasping video, and output a corrected action. The ground truth action for each frame is taken from the corresponding source dataset. During training, we randomly select 5 frames from around the key grasp-time in each generated video for the model to predict an action sequence. To optimize resource usage, we employ a complex QA pair structure that allows for joint training of the VLA-in-Loop model (in its IDM role) and the discriminator model (in its evaluator role). The model takes an observation sequence $O_{t':t'+T}$ and a text prompt $L$ as input and outputs an action sequence $a_{t':t'+T}$, with a default action horizon of T=5.

Table 1: Comparison on SIMPLER with WidowX and Google Robot. $^\dagger$ denote the method fine-tuned in domain. Best results are in bold.

| Method | SIMPLER with WidowX | | | | | SIMPLER with Google Robot | | | | | | | | | |
|---|---|---|---|---|---|---|---|---|---|---|---|---|---|---|---|
| | Tasks (Grasp/Success %) | | | | Avg. (G/S) | Visual Matching (Success %) | | | | | Variant Aggregation (Success %) | | | | |
| | Spoon on towel | Carrot on plate | Block on block | Put egg. on basket | | Pick Coke | Move Near | O/C Drawer | Top Dr. Apple | Avg. | Pick Coke | Move Near | O/C Drawer | Top Dr. Apple | Avg. |
| RT-1$^\dagger$ | — | — | — | — | — | 85.7 | 44.2 | 73.0 | 6.5 | 52.4 | 89.8 | 50.0 | 32.3 | 2.6 | 43.7 |
| RT-1-X | 16.7/0.0 | 20.8/4.2 | 8.3/0.0 | 0.0/0.0 | 11.5/1.1 | 56.7 | 31.7 | 59.7 | 21.3 | 42.4 | 49.0 | 32.3 | 29.4 | 10.1 | 30.2 |
| RT-2-X | — | — | — | — | — | 78.7 | 77.9 | 25.0 | 3.7 | 46.3 | 82.3 | 79.2 | 35.3 | 20.6 | 54.4 |
| Octo-Base | 34.7/12.5 | 52.8/8.3 | 31.9/8.3 | 66.7/43.1 | 46.5/16.0 | 17.0 | 4.2 | 22.7 | 0.0 | 11.0 | 0.6 | 3.1 | 1.1 | 0.0 | 1.2 |
| Octo-Small | 77.8/47.2 | 27.8/9.7 | 40.3/4.2 | 87.5/56.9 | 58.4/29.5 | — | — | — | — | — | — | — | — | — | — |
| OpenVLA | 4.1/0.0 | 33.3/0.0 | 12.5/0.0 | 8.3/4.1 | 14.6/1.0 | 18.0 | 56.3 | 63.0 | 0.0 | 34.3 | 60.8 | 67.7 | 28.8 | 0.0 | 39.3 |
| RoboVLMs$^\dagger$ | 70.8/45.8 | 33.3/20.8 | 54.2/4.2 | 91.7/79.2 | 62.5/37.5 | 77.3 | 61.7 | 43.5 | 24.1 | 41.9 | — | — | — | — | — |
| SpatialVLA | 25.0/20.8 | 41.7/20.8 | 58.3/25.0 | 79.2/70.8 | 51.2/34.4 | 81.0 | 69.6 | 59.3 | — | 71.9 | 89.5 | 72.7 | 41.8 | — | 68.8 |
| SpatialVLA$^\dagger$ | 20.8/16.7 | 29.2/25.0 | 62.5/29.2 | **100.0/100.0** | 53.1/42.7 | 86.0 | 77.9 | 57.4 | — | 75.1 | 88.0 | 72.7 | 41.8 | — | 70.0 |
| ThinkAct$^\dagger$ | —/58.3 | —/37.5 | —/8.7 | –/70.8 | —/43.8 | 92.0 | 72.4 | 50.0 | — | — | 84.0 | 63.8 | 47.6 | — | — |
| *CogAct* | —/71.7 | —/50.8 | —/15.0 | -/67.5 | —/51.8 | 91.3 | 85.0 | 71.8 | 50.9 | 74.8 | 89.6 | 80.8 | 28.3 | 46.6 | 61.3 |
| **VLA-in-Loop** | **83.3/75.0** | **62.5/58.3** | **62.5/37.5** | 87.5/83.3 | **74.0/63.5** | **94.7** | **92.3** | **74.5** | **52.6** | **78.5** | **91.8** | **86.3** | 38.5 | 51.3 | 67.0 |

## 4.2 EVALUATION PERFORMANCE

To comprehensively validate the effectiveness, efficiency, and real-world applicability of the VLA-in-Loop framework, we conducted a series of extensive experiments in both standardized simulation environments and on a physical robot platform. The visualization of processing is shown in Figure 4.

**Evaluation on Simulation** In SIMPLER (Li et al., 2024c), we test our framework on two commonly used robotic platforms: WidowX and Google Robot. As shown in Table 1, *VLA-in-Loop* demonstrates superior performance on the SIMPLER benchmark with the WidowX arm. Our method achieves an average grasp and success rate of 63.5%, significantly outperforming the next best method, CogAct (Li et al., 2024a) (51.8%). Notably, our method excels in tasks requiring both precise grasping and spatial reasoning, such as "Put spoon on towel" (83.3% success) and "Stack green block on yellow block" (62.5% success), where many other methods fail completely. This highlights the effectiveness of our online correction mechanism in fine-grained manipulation where base policies often suffer from accumulated errors.

When evaluated with the Google Robot, *VLA-in-Loop* continues to show strong performance. In the "Visual Matching" category, our method achieves an average success rate of 78.5%, surpassing all other methods,

including RT-1 (O'Neill et al., 2024), which was fine-tuned on Google Fractal dataset (Brohan et al., 2022), and specialized models like CogAct Li et al. (2024a). In the more challenging "Variant Aggregation" tasks, VLA-in-Loop also obtains the top average score of 67.0%. The results underscore our model's ability to adapt its fine-grained correction strategy to different robot kinematics and dynamics, maintaining high performance without requiring in-domain fine-tuning.

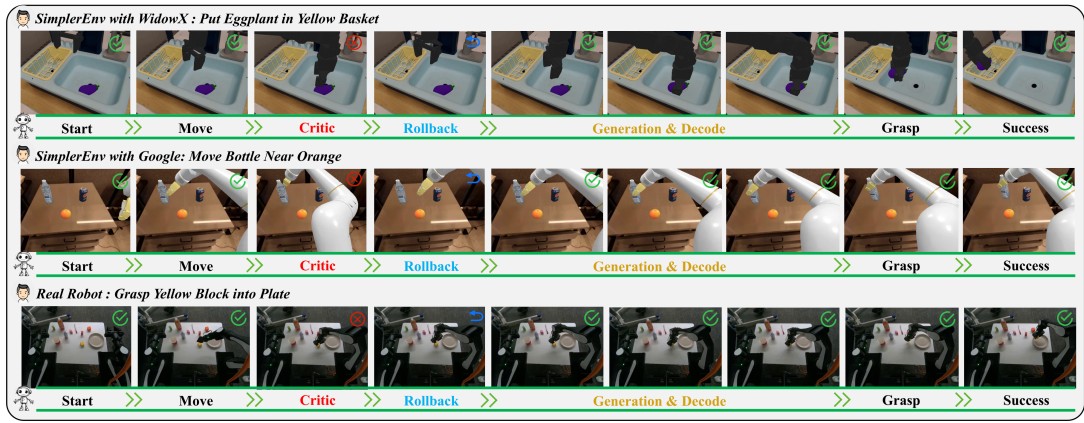

Figure 4: Qualitative examples of our failure correction process. Each row shows a sequence: (a) the initial failed grasp proposed by the base VLA, (b) the rolled-back state, (c) a keyframe from the imagined successful video generated by our world model, and (d) the final successful grasp executed by our framework. Examples are shown for (i) WidowX and (ii) Google Robot in simulation, and the (iii) Xiaomi Robot in the real world.

**Real-World Set Up** The benefits of our correction framework are even more pronounced in the physical world, where unmodelled dynamics and visual noise are prevalent. Our method not only improves the grasp itself but also positively impacts the overall task success, indicating a more robust trajectory following the correction. As summarized in Table 2, we evaluated our framework on a series of challenging real-world tasks and compared it against the baseline VLA policy. The performance gap is particularly significant in tasks requiring high precision under uncertainty, such as "Grasp Watercup" (a 16.7% improvement in success) and "Grasp Block into Plate" (a 17.3% improvement). The difference between the "Grasp" and "Success" metrics is also revealing: even when the baseline manages a grasp, it may be unstable, leading to subsequent failure. Our method's higher final success rate indicates that the imagined correction provides a better-quality grasp, setting the robot up for a more stable and successful completion of the entire task.

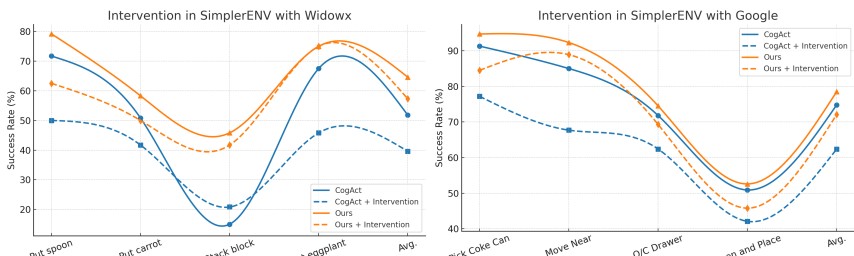

Figure 5: Robustness to online perturbations in simulation environments.

### 4.3 PROCESSING OF FAILURE PREDICTION AND CORRECTION

We demonstrate our method's efficacy through qualitative failure correction examples and quantitative robustness tests. Figure 4 illustrates our end-to-end correction pipeline: a failed grasp from the base VLA is identified (Critic), the state is rolled back, successful image frames are imagined by the world model (Generation), and a corrected action (Decode parse) is successfully executed. To evaluate robustness, we introduced controlled perturbations during tasks (Figure 5). Unlike baselines that either fail or simply halt, our framework maintains a significantly higher success rate under increasing interference. This highlights our framework's superior ability to handle unexpected online disturbances through proactive, generative correction rather than just passive failure detection.

Table 2: Real-world results on Different Tasks.

| Setting | Grasp Watercup | Grasp Block into Plate | Move Block Near | Stack Block | Avg. |
|---|---|---|---|---|---|
| Grasp (%). | | | | | |
| Baseline | 66.7 | 75.0 | 75.0 | 66.7 | 70.9 |
| Ours | 79.2 | 91.7 | 87.5 | 83.3 | 85.4 |
| Success (%). | | | | | |
| Baseline | 54.2 | 58.3 | 75.0 | 66.7 | 63.6 |
| Ours | 70.9 | 75.0 | 83.3 | 75.0 | 76.1 |

Table 3: Impact of generative and rollback depth.

| Setting | Put spoon on towel | Put carrot on plate | Stack green on yellow | Put eggplant on yellow | Avg. |
|---|---|---|---|---|---|
| Success (%). | | | | | |
| Baseline | 71.7 | 50.8 | 15.0 | 67.5 | 51.8 |
| Correction w/o Rollback | 54.2 | 54.2 | 45.8 | 70.8 | 56.3 |
| Correction with Rollback | | | | | |
| *Rollback* $5 \times steps$ | 58.3 | 50.0 | 33.3 | 75.0 | 54.2 |
| *Rollback* $10 \times steps$ | 66.7 | 58.3 | 41.7 | 79.2 | 61.5 |
| *Rollback* $15 \times steps$ | 62.5 | 50.0 | 41.7 | 83.3 | 59.4 |
| *Rollback* $20 \times steps$ | 75.0 | 58.3 | 37.5 | 83.3 | 63.5 |

### 4.4 ABLATION STUDY

To dissect the contributions of our framework's core components, we conducted a detailed ablation study on four challenging tasks. The results, presented in Table 3, systematically evaluate the impact of the generative correction and the state rollback mechanism. Our analysis reveals two key insights. First, we evaluate the Baseline VLA policy as the performance floor. In the Correction w/o Rollback setting, the supervisor attempts to correct the policy from the immediate state where failure is predicted. We observe that this approach provides only marginal or inconsistent benefits. This suggests that by the time a failure is imminent, the state may already be irrecoverable, limiting the effectiveness of any correction attempted from that point. Second, we analyze the effect of Correction with Rollback at varying depths (5, 10, 15, and 20 steps). The results show a clear and positive trend: increasing the rollback depth generally improves task success rates. By reverting to an earlier, "healthier" state, the generative model is provided a more viable starting point to imagine a successful trajectory. This confirms that the state rollback is a critical component of our framework, enabling the agent to effectively escape near-failure states and replan a successful course of action. The performance improvements begin to plateau at a certain depth, indicating a trade-off between finding a recoverable state and maintaining task progress.

## 5 CONCLUSION AND LIMITATION

We introduced VLA-in-the-Loop, a framework that enhances VLA robustness through on-demand, online correction. Our key contribution is a novel, efficient use of a composite world model that combines discriminative reasoning and generative imagination to correct policies at critical moments. This approach paves the way for more reliable and adaptable robotic agents. Future work could focus on reducing the latency of the generative model and improving the physical plausibility of imagined trajectories. We also plan to extend the framework to a broader set of tasks and explore methods for the agent to learn when to trigger an intervention. Ultimately, our paradigm moves towards robotic systems that intelligently self-correct by synergizing large-scale policies with the on-demand reasoning capabilities of foundation models.

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

# Appendix A

# Technical Appendices and Supplementary Material

In this section, we provide supplementary material to support the findings presented in the main paper, including an extended dataset and training information, Appendix A.1, detailed network architectures and data processing Appendix A.2, additional experimental results, Appendix A.3, the intervention design of Real-world and simulation robot Appendix A.4, and qualitative visualizations on various tasks, Appendix A.5. Statement of using LLM tools is present at Appendix A.6

## A.1 TRAINING DATASET DETAIL

Our datasets are sourced from three main sources: the BridgeV2 dataset (Walke et al., 2023), the Google Robotics dataset (Brohan et al., 2022), real-world data collected using the ALOHA robotic platform and data from the Xiaomi wheeled robot.

### A.1.1 OPENSOURCE DATASETS

**BridgeV2 dataset** The BridgeV2 dataset serves as the primary foundation for our training data. It is a large-scale publicly available robotics data set that includes various manipulation tasks, including grasping, picking, and placing objects in varied environments. Comprising more than 100,000 video clips and millions of frames collected from multiple robotic platforms, BridgeV2 provides rich multimodal data, such as RGB images, depth information, and action trajectories, which are essential for training vision-language-action models. In our work, we leverage the subsets of BridgeV2 for both the discriminator and the generator, enabling the world model to learn from simulated and diverse failure-success pairs without extensive real-world collection.

**Google Fractal dataset** The Google Robotics Dataset serves as the foundational corpus for our model training. It is a large-scale, in-house robotics dataset encompassing over 700 distinct, everyday manipulation tasks guided by natural language instructions. Comprising more than 130,000 demonstration episodes collected from a fleet of 13 mobile robotic manipulators, the dataset provides rich multimodal data streams, such as RGB image observations paired with corresponding action commands and language-based goals. This data is critical for training generalist Vision-Language-Action (VLA) models designed for broad generalization across novel tasks, objects, and environments. In our work, we utilize this dataset to train a general-purpose, instruction-following agent, enabling it to compose and extrapolate from learned skills to address previously unseen long-horizon problems.

### A.1.2 DATA FOR REAL ROBOT EXPERIMENTS

**ALOHA Hardware Setup** Our ALOHA hardware platform Figure 1a is based on the Tsinghua Airbot system. It consists of two 6-DoF lightweight manipulators mounted on a compact mobile base, each equipped with a 1-DoF parallel gripper and a wrist-mounted RGB-D camera for close-range manipulation perception. An Intel RealSense D435 RGB-D camera is mounted on an adjustable overhead mast and streams 640×480 RGB-D frames at 30 Hz. Whenever the overhead camera pose changes, we perform hand-eye calibration using an AprilTag calibration board to estimate the rigid transformation from the robot base to the camera. All teleoperated trajectories are transformed into the overhead camera coordinate frame during preprocessing, while actions predicted by the learned policy are mapped back to the robot base frame using the calibrated extrinsics. Cartesian incremental commands output by the model are integrated into target end-effector poses and converted to joint commands via a damped least-squares inverse kinematics solver, with joint velocity and acceleration limits enforced.

**Xiaomi Robot Setup**   Figure 1b illustrates our 7-DoF Xiaomi Robot setup. The head integrates an Intel RealSense D435 RGB-D camera to provide close-range task observations, streaming 640×480 RGB-D frames at 30 Hz. This setup enables real-time perception of the workspace, including object detection and pose estimation. All policy outputs are Cartesian coordinate increments expressed in the robot base frame, allowing for direct integration with the arm's control system via a ROS-based interface. Data are collected at 10 Hz using teleoperation, ensuring compatibility with low-latency applications.

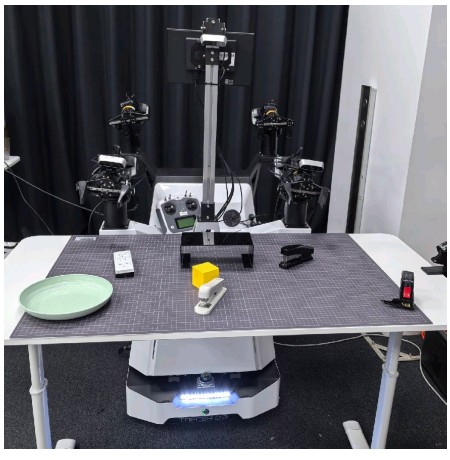
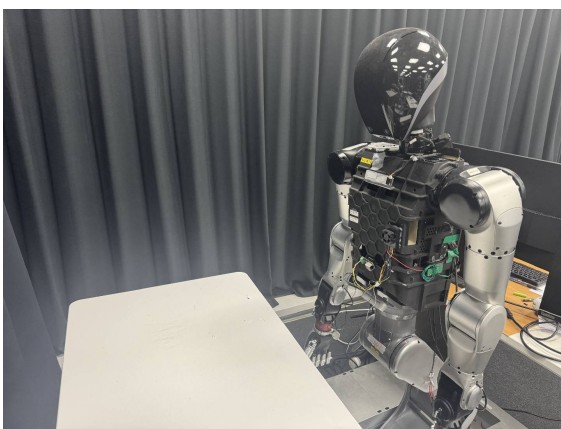

(a) ALOHA Hardware setup                         (b) Xiaomi robot setup

Figure 1: Different robot setups in real-world deployment.

**Data PreProcessing**   We collect data using a teleoperated robotic arm controlled by a SpaceMouse, capturing trajectories that include joint positions, end-effector poses, and RGB-D images. The raw data undergo several preprocessing steps. First, we filter out noisy frames by applying a median filter with a kernel size of 5 and removing outliers beyond a threshold of 3 standard deviations. Next, we synchronize the multimodal data streams and downsample the ALOHA data from 30Hz to 10Hz to match the Xiaomi robot data, thereby ensuring temporal consistency across datasets. Finally, we convert the processed data to the RLDS format, aligning it with the BridgeV2 dataset structure, which includes standardized fields for observations, actions, and rewards.

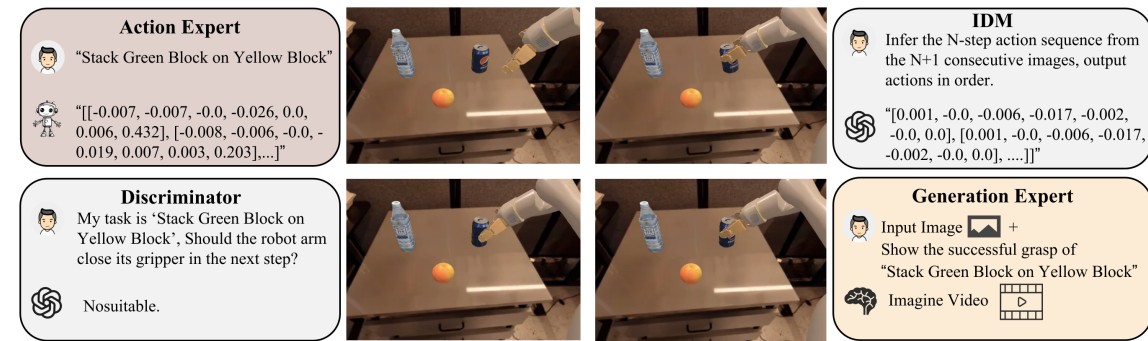

Figure 2: Prompt processing of our framework.

## A.2 MODEL AND TRAINING DETAILS

### A.2.1 BASE VLA POLICY

In the Base VLA Policy, we selected CogACT (Li et al., 2024a) as the foundational model, which is a state-of-the-art VLA model. Its innovative cognition-action synergy architecture efficiently separates the cognitive extraction capabilities of the VLM from the action prediction module, significantly enhancing the generalization and robustness of robots in complex manipulation tasks.

**Architecture** CogACT is an integrated VLA framework that maps multimodal observations to structured action sequences. Given an input image $I_t$, a hybrid visual encoder (DINOv2 + SigLIP) extracts patch-level features, which are linearly projected into the language model's embedding space to form visual tokens $\mathcal{V} = \{v_1, v_2, \ldots, v_{N_\mathcal{V}}\}$. In parallel, textual inputs are tokenized to $\mathcal{L} = \{l_1, l_2, \ldots, l_{N_\mathcal{L}}\}$ and embedded in $\mathbb{R}^d$ by a Llama2-based autoregressive Transformer.

Multimodal fusion occurs inside the Transformer stack by jointly attending over $\mathcal{V}$ and $\mathcal{L}$, yielding a compact cognition token $\mathcal{Z} \in \mathbb{R}^d$ that summarizes task-relevant scene semantics and instruction intent. This token conditions a Diffusion Transformer (DiT) responsible for action synthesis. The DiT performs iterative denoising over latent action trajectories, exploiting long-range temporal dependencies through its Transformer backbone.

During training, ground-truth future action segments $x_0$ (covering the next $n$ steps) are perturbed via the forward diffusion process:

$$x_t = \sqrt{\alpha_t} \cdot x_0 + \sqrt{1 - \alpha_t} \cdot \epsilon, \tag{3}$$

where $\alpha_t \in (0, 1)$ regulates the noise level at timestep $t$, and $\epsilon$ is standard Gaussian noise. The DiT is optimized to predict the injected noise conditioned on the noisy sample $x_t$, the timestep $t$, and the cognition token $\mathcal{Z}$:

$$\text{DiT}(x_t, t, \mathcal{Z}) \approx \epsilon. \tag{4}$$

This end-to-end design enables coherent grounding from perception and language to executable control, improving both sample efficiency and action reliability in robotic tasks.

**Deployment** For efficient inference in robotic tasks, CogACT employs several hyperparameters tuned for real-time action generation. The diffusion process uses 50–100 steps to iteratively denoise action trajectories, balancing precision and computational cost. The sampling temperature is set to 1.0 by default and is adjustable to 0.7–0.9 for deterministic outputs. Action prediction spans 8–16 future steps, enhanced by an Adaptive Action Ensemble with 3–5 samples and a cosine similarity threshold of 0.8 for consistency. The batch size is typically 1 for low-latency deployment, while noise scheduling follows a cosine scheme aligned with training. These settings ensure reliable performance across diverse manipulation scenarios.

### A.2.2 WORLD-MODEL-DISCRIMINATOR

Our discriminator is built upon Qwen-VL 2.5 (Bai et al., 2025), a state-of-the-art vision-language model optimized for multimodal understanding and evaluation. It functions as the primary component for assessing action-related judgments, specifically determining gripper states by analyzing visual inputs alongside task-specific instructions. When integrated with the base VLA policy, the discriminator improves the system's ability to optimize context-aware action predictions. This refinement leads to more reliable performance in object manipulation tasks.

**Architecture** The Qwen-VL 2.5 architecture combines a vision encoder with a LLM backbone, enabling seamless processing of image-text pairs. Given an input image $I$ and a textual prompt, the vision encoder extracts high-level features, which are fused with tokenized text embeddings in the LLM's attention layers.

**Discriminator**

**System:** You are a robot agent. Your task is to determine whether the current state of the robotic arm is suitable for grasping or releasing, and you can act as an IDM to decode the corresponding action sequence based on the given video.

**Query** : My task is 'Put spoon on towel'. Should the robot arm close its gripper in the next step? Your response should only be  'suitable ' or 'nosuitable '."

**Answer** : Suitable.

**Query** :Infer the 4-step action sequence from the 5 consecutive images, output actions in order. Output format requirement:
   - The result must be a JSON array with exactly 4 elements.
   - Each element must be an array of 7 floating point numbers.
   - Do not add explanations, text, or formatting outside the JSON array."

**Answer :** [[0.001, -0.002, -0.005, -0.016, -0.003, -0.013, 1.0], [0.001, 0.0, -0.013, -0.035, -0.011, -0.03, 1.0], [0.001, -0.003, -0.011, -0.032, -0.002, -0.006, 1.0], [0.001, -0.0, -0.006, -0.017, -0.002, -0.0, 0.0]]

**Action Decoder**

Figure 3: Unified QA pairs of shared Discriminator and VLA-in-Loop Model.

This multimodal fusion allows the model to reason over visual scenes and generate binary responses for action queries.

**Training Details**   In our setup, the generator is fine-tuned to focus on gripper state decisions. For a given image where the base VLA model detects a potential gripper closure event, we construct a prompt template that incorporates the task description and the gripper state query. The model outputs a concise response, effectively acting as a binary classifier conditioned on visual and linguistic cues. This design leverages Qwen-VL's pre-trained capabilities for efficient adaptation to robotic tasks without requiring extensive architectural modifications. Hyperparameters for fine-tuning are detailed in  Table 1 (shared with the discriminator for consistency), with adjustments for VL-specific tasks, such as a maximum sequence length of 2048 to accommodate image tokens. Training emphasizes low-latency inference, with deployment on edge devices for real-time robotic control.

**Data Construction**   The training dataset for the discriminator was constructed by labeling critical keyframes from both the BridgeV2 dataset and our real-robot collections. We curated a total of 102k frames, with 100k from a BridgeV2 subset and 2k from real-world trials. Each frame was manually or semi-automatically labeled as 'suitable' or 'unsuitable' for grasping, ensuring a balanced 1:1 ratio of positive to negative samples to prevent classification bias. An 'unsuitable' state was defined by trajectories where the gripper was not intended to close or where the pre-grasp pose was geometrically infeasible. Training is performed using question-answer (Q&A) pairs derived from the BridgeV2 and real-robot datasets. Each training example consists of an image-text pair, where the image depicts a manipulation scene (e.g., a robot

arm approaching a blue block on a yellow block), and the text is a structured prompt: *<image> My task is 'put the blue block on the yellow block'. Should the robot arm gripper_state its gripper in the next step ? Your response should only be 'suitable' or 'unsuitable'*, as shown in Figure 3.

### A.2.3 WORLD-MODEL-GENERATOR

Our generative world model is built upon the WAN2.1 (Wan, 2025) architecture, a state-of-the-art video generation model optimized for short-form, goal-conditioned video synthesis. It serves as the imaginative component of our framework, tasked with creating plausible future video sequences that depict successful task completion.

**Architecture** The WAN2.1 model employs a latent diffusion architecture. An input image I is first passed through a vision encoder to produce a compact latent representation. This image latent, along with text embeddings from a prompt, are used to condition a diffusion process operating in the latent space. The model is trained to denoise a random latent tensor over several timesteps, progressively refining it into a latent representation of a future video sequence. A final video decoder then translates this latent sequence back into a pixel-space video. This latent-space operation allows for efficient generation of high-resolution video clips.

**Training Details** We fine-tune the pre-trained WAN2.1 model on a specialized dataset of successful robotic grasps. When the discriminator identifies a need for correction from a rolled-back state $o_{t-k}$, the generator is provided with this image and a task-specific prompt (e.g., *<image> My task is to 'pick up the red block'. Generate a video of the robot arm successfully grasping the red block.*). The model is trained to output a short video (5-20 frames) that starts from $o_{t-k}$ and ends with a successful grasp. The training objective is to minimize the reconstruction error between the generated video and ground-truth successful video clips from our datasets (BridgeV2 and real-robot data), ensuring the generated motions are physically plausible and task-relevant. Hyperparameters for fine-tuning are detailed in Table 2.

**Data Construction** The training corpus for the generative model was sourced from both large-scale public datasets and targeted real-world collection. The primary dataset consists of 33,000 short video clips from the BridgeV2 dataset depicting a wide variety of successful manipulation behaviors. To adapt the model to our specific robotic platforms and environments, we supplemented this with a fine-tuning dataset of 200 successful grasping videos (100 each from the Xiaomi Robot and ALOHA). Each collected video is between 5 and 20 frames long, capturing the crucial moments of a successful grasp.

### A.2.4 VLA-IN-LOOP MODEL

The VLA-in-Loop model is the unified agent that performs both failure prediction and action correction. It is built upon the Qwen-VL 2.5 architecture, leveraging its powerful multimodal understanding capabilities to serve these dual roles efficiently.

**Architecture** The Qwen-VL 2.5 architecture comprises a vision encoder and an LLM backbone. When processing input, the vision encoder extracts high-level features from images or video frames. These visual tokens are then seamlessly integrated with tokenized text embeddings within the LLM's attention layers. This deep fusion allows the model to reason jointly over complex visual scenes and natural language queries, making it highly suitable for our multi-task QA formulation for both evaluating and generating robotic actions.

**Training Details** The unified model is fine-tuned using a multi-task Question-Answering (QA) scheme. Its training data consists of two distinct types of QA pairs, as shown in Figure 3:

- **Evaluator QA Pairs**: These are used for failure prediction. The input is a single image $o_t$ and a prompt asking for a "suitable" or "unsuitable" judgment on a proposed action.
- **Actor QA Pairs**: These are used for action correction. The input consists of the imagined successful video $v_s uccess$ concatenated with the current observation $o_t$, and a prompt asking for the corrected action: *<video> Infer the 4-step action sequence from the 5 consecutive images, output actions in order*. The model is trained to output the precise 7-DoF action tuple corresponding to the ground-truth action $a_t^*$ from the demonstration data.

By training on these complex, mixed QA pairs, the model learns a shared representation that is grounded in both high-level semantic judgment and low-level action parameterization, which is critical to the framework's overall performance. Hyperparameters for fine-tuning are shared with the discriminator for consistency, with adjustments for action-specific tasks, such as a maximum sequence length of 2048 to accommodate video tokens.

**Data Construction**    The dataset for the unified VLA-in-Loop model is structured entirely in a Question-Answering (QA) format to support its dual roles. For its function as an evaluator, we use the same 'suitable'/'unsuitable' labeled frames from the discriminator's dataset, formatted as VQA pairs. For its function as an actor, we create a new set of QA pairs. For each instance where a correction is needed, we take the imagined successful video $v_s uccess$ generated by our world model and pair it with a prompt asking for the correct action sequence. The ground-truth action $a_t^*$ is sourced from the original successful demonstration in the dataset that corresponds to the initial failed state

<table>
<tr><td colspan="2">Table 1: Discriminator Hyperparameters</td><td colspan="2">Table 2: WAN2.1 Training Hyperparameters</td></tr>
<tr><td>Hyperparameter</td><td>Value</td><td>Hyperparameter</td><td>Value</td></tr>
<tr><td>Number of Train Epochs</td><td>15</td><td>Height</td><td>256</td></tr>
<tr><td>Per Device Train/Eval Batch Size</td><td>1</td><td>Width</td><td>256</td></tr>
<tr><td>Train Type</td><td>Full</td><td>Dataset Repeat</td><td>2</td></tr>
<tr><td>Global Batch Size</td><td>28</td><td>Num Epochs</td><td>5</td></tr>
<tr><td>Max Length</td><td>2048</td><td>Gradient Accumulation Steps</td><td>1</td></tr>
<tr><td>Learning Rate</td><td>2e-5</td><td>Learning Rate</td><td>1e-4</td></tr>
<tr><td>Warmup Ratio</td><td>0.1</td><td>LoRA Base Model</td><td>dit</td></tr>
<tr><td>Weight Decay</td><td>0.05</td><td>LoRA Target Modules</td><td>q,k,v,o,ffn.0,ffn.2</td></tr>
<tr><td>Gradient Accumulation Steps</td><td>4</td><td>LoRA Rank</td><td>32</td></tr>
</table>

## A.3    MORE EXPERIMENTS

**Model size VS. Latency**    We performed a comprehensive ablation study across different model configurations (Setting-1, Setting-2) to identify the optimal operating point, shown as in Table 3.

- For setting-1, we utilize a 3B Discriminator (0.3s) and a 5B Generator (4s), achieving a +11.7% success rate improvement on WidowX (63.5% vs. 51.8% baseline) while keeping the intervention cost at 5 seconds with incurred in 21.5% of episodes; we consider this the optimal balance for real-time deployment.
- For setting-2, scaling to 7B/14B models boosts success by +21.1% but increases latency to 21s. Although this setting demonstrates the strong scaling potential of our VLA-in-the-Loop framework (i.e., better foundation models yield better policies), it is less practical for real-time use. This justifies our choice of Setting-1 for the main paper.

Although larger models demonstrate strong scaling potential, we consider Setting-1 to be the optimal balance for real-time deployment.

Table 3: Metrics under different model size of discriminator and generator.

| | Discriminator(QwenVL-2.5) | | Generator (WAN) | |
|---|---|---|---|---|
| Params | 3B | 7B | 5B(V2.2) | 14B(V2.1) |
| Latency | 0.3s | 0.9s | 4s | 21s |
| Positive Rate | 90.1% | 93.1% | \ | \ |
| | **WidowX** | **Google-VM** | **Google-VA** | **Real robot** |
| Base policy (CogAct) | 51.8% | 74.8% | 61.3% | 63.6% |
| Setting-1 | 3B | | 5B | |
| | 63.5%(**+11.7**) | 78.5%(**+3.7**) | 67.0%(**+5.7**) | 76.1%(**+12.5**) |
| Setting-1 Intervention(max=2) | 65.6%(**+13.8**) | 78.7%(**+3.9**) | 68.1%(**+6.8**) | 78.1%(**+14.5**) |
| Setting-2 (Large Model size) | 7B | | 14B | |
| | 72.9%(**+21.1**) | 78.9%(**+4.1**) | 68.6%(**+7.3**) | 81.3%(**+17.7**) |

**Isolating the Value of Imagination**  To isolate the specific contribution of the generated video, we conducted a controlled experiment as Table 4. Since our base policy is deterministic, utilizing the Discriminator solely to trigger a rollback results in a deadlock: the robot reverts to the previous state (), receives the exact same observation, and repeats the identical failing action sequence. Therefore, to construct a functional "No-Generation" baseline, we introduced action noise to break this loop. Instead of guiding the robot with an imagined video after rollback, we injected Gaussian noise into the predicted actions of the base policy to force a trajectory deviation. Results (Value of Imagination): As shown in the table above, simply perturbing the policy ("Noise Variant") offers no benefit, whereas our Generative approach yields significant gains. The "No-Generation" variant (52.1%) fails to improve upon the baseline (51.8%), confirming that rollback alone is insufficient. The significant performance leap to 63.5% is directly attributable to the World Model's imagined video, which provides the specific, dense visual guidance necessary to correct the trajectory.

Table 4: Ablation study on world model interventions.

| | Avg. | Spoon on Towel | Carrot on Plate | Stack Cube | Put Eggplant in Basket |
|---|---|---|---|---|---|
| CogACT | 51.8 | 71.7 | 50.8 | 15.0 | 67.5 |
| CogACT + Noise | 52.1 | 70.8 | 50 | 16.7 | 70.8 |
| CogACT + Ours | 63.5 | 75.0 | 58.3 | 37.5 | 83.3 |

**Evaluation on LIBERO**  The LIBERO benchmark (Liu et al., 2023) tests long-horizon skills. As shown in Table 3, *VLA-in-Loop* achieves a state-of-the-art average success rate of 87.1%. It particularly excels in the "Long" horizon category with a score of 82.0%, significantly outperforming the next best method, ThinkAct (70.9%). This demonstrates that our keyframe-triggered correction approach is not only effective for short-horizon grasps but also provides crucial stability for long-term tasks, preventing early failures and ensuring the policy remains on a successful trajectory.

**Time-consuming.**  Our event-triggered design creates a favorable trade-off between latency and robustness. The significant time cost of intervention is only paid when necessary, maintaining real-time perfor-

Table 5: Comparison results on LIBERO with Franka

| Method | Goal | Spatial | Object | Long | Average |
|---|---|---|---|---|---|
| Diffusion (Chi et al., 2023) | 68.3 | 78.3 | 82.5 | 50.5 | 72.4 |
| Octo-Base (Team et al., 2024) | 84.6 | 78.9 | 85.7 | 51.1 | 75.1 |
| OpenVLA (Kim et al., 2024) | 79.2 | 84.7 | 88.4 | 53.7 | 76.5 |
| TravceVLA (Zheng et al., 2024) | 75.1 | 84.6 | 85.2 | 54.1 | 74.8 |
| SpatialVLA (Qu et al., 2025) | 78.6 | 88.2 | 89.9 | 55.5 | 78.1 |
| ThinkAct (Huang et al., 2025) | 87.1 | **88.3** | 91.4 | 70.9 | 84.4 |
| Cot-VLA (Zhao et al., 2025) | **87.6** | 87.5 | 91.6 | 69.0 | 81.1 |
| **VLA-in-Loop** | 87.1 | 87.8 | **91.6** | **82.0** | **87.1** |

mance for the majority of the task while drastically improving reliability at critical moments. Specifically, a standard step with the base VLA policy takes approximately 0.2 seconds. When our VLA-in-Loop correction is triggered at a keyframe, the entire process is significantly longer, dominated by the video generation. The discriminator requires 0.95 seconds to evaluate the impending action, and the generative model takes approximately 20 seconds to create the corrective video.

While a 4 or 21-second (presented in Table 3) pause for correction is substantial, it is a strategic and acceptable trade-off. This intervention is sparse, occurring only when the highly accurate discriminator predicts a critical failure, which would ultimately be far more time-consuming, often requiring a full task reset or human intervention. Therefore, our framework is optimized for overall task success and autonomy, rather than the latency of any single corrective step.

### A.4 INTERVENTION DETAILS.

To rigorously evaluate the robustness of our framework, we introduced controlled, online perturbations during task execution in both real-world and simulated settings. These interventions were designed to test the policy's ability to adapt to unexpected changes that deviate from the training distribution.

**Real-World Perturbations.** In the real-world experiments with the Xiaomi Robot and ALOHA platforms, perturbations were applied manually but consistently. During a task execution, either the target object itself would be moved to a new position, or the container/surface on which the object was placed (e.g., a tray or a table region) would be shifted. These movements were performed while the robot was in motion, forcing the policy to dynamically re-evaluate and correct its trajectory in real-time.

**Simulation Perturbations** For the simulation experiments in SIMPLER and LIBERO, we introduced noise directly into the robot's control space to simulate actuator imprecision or external physical forces. At random intervals, we added zero-mean Gaussian noise to the predicted end-effector action. Specifically, for a given action $a_t = (xyz, rpy, g\_t)$, the executed action $a'_t$ would be $(xyz + \epsilon_p, rpy + \epsilon_o, g\_t)$, where $\epsilon_p$ and $\epsilon_o$ are noise vectors sampled from $N(0, \sigma^2)$. Here, we apply the disturbed variance $\sigma^2$ within the range of [0.001, 0.01] to create different levels of perturbation intensity, allowing us to analyze the policy's performance degradation under increasing levels of noise.

### A.5 VISUALIZATIONS

To provide a clearer understanding of the challenges addressed in our experiments, we include a series of visualizations for the tasks performed, as shown in Figure 4. These images showcase the diversity of our evaluation suite, spanning both simulated and real-world environments. The figures display key moments

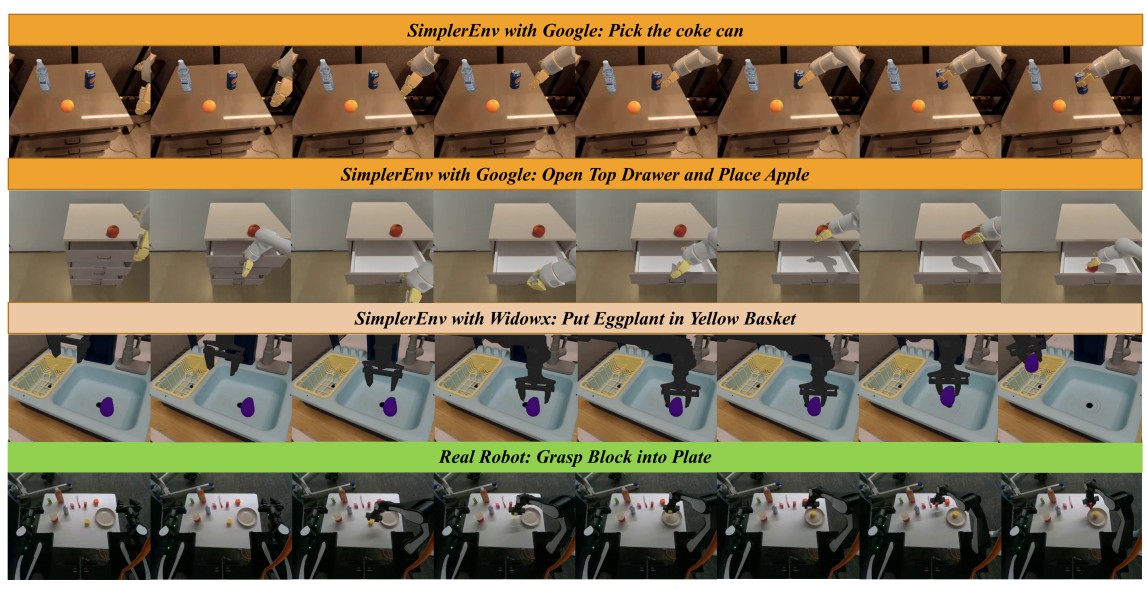

Figure 4: Visualization of various tasks.

from various tasks performed on the SIMPLER platform, as well as on the physical Xiaomi Robot and ALOHA systems. This visual evidence highlights the complexity of the object arrangements, the required precision for manipulation, and the dynamic conditions under which our VLA-in-Loop framework was tested.

## A.6 STATEMENT

***LLM Assistance: We employed LLMs exclusively for text polishing; they were in no way involved in research ideation.***

## A.7 ETHICS STATEMENT

We acknowledge and adhere to the ICLR Code of Ethics throughout this work. Our research does not involve human subjects or sensitive personal data, and all experiments were conducted in compliance with ethical research practices.

## A.8 REPRODUCIBILITY STATEMENT

We have made extensive efforts to ensure reproducibility. In the supplementary materials, we provide details of our training datasets and training detail in Appx.A.1 . We further include hyperparameters in Appx.A.2.

