# OpenReview forum: "VLA-IN-THE-LOOP: ONLINE POLICY CORRECTION WITH WORLD MODELS FOR ROBUST ROBOTIC GRASPING"
_ICLR.cc/2026/Conference — Submitted to ICLR 2026_

### Official Review · Reviewer_eohM · 2025-10-24

**Soundness:** 3
**Presentation:** 3
**Contribution:** 2
**Rating:** 4
**Confidence:** 4

**Summary:**

The paper presents a way of coupling a VLA model with a video generation model, so that the latter "oversees" the former. Specifically, for important actions, like grasping, a discriminator (qwen-based VLM) detects if the VLA action is likely to succeed. If not, the actions are rolled back, and a video model generates a successful trajectory while an IDM model extracts the actions from this trajectory. This new action sequence is used instead. The authors show that this improves the baseline performance both in sim and real.

**Strengths:**

- The idea is simple and well explained; a reader gets the gist quickly.
- The authors show good improvements both in real and sim

**Weaknesses:**

- The novelty is limited
- The keyframe detection seems to only be grasping. This is a somewhat "hardcoded" method, which isn't very general
- It is not clear that "rollback" is a generally feasible strategy in the real world
- Missing baseline/citation to DreamGen

**Questions:**

- Rollback isn't always feasible in the real world. E.g., the robot might drop something on the floor, and you can't rollback by inverting gravity. How does your method work in these settings?
- From table 3, it seems that more rollback works better. Is not not possible to roll back to the start of the episode and let the video model+IDM do the complete action generation? And following your table, isn't that likely to work better than your proposed method?
- Is there a way to generalize your method beyond just working for grasping?

---

> ### Author Response · Authors · 2025-11-24
> **Response to Reviewer eohM, (1/2)**
>
> We sincerely thank the reviewers for their professional and constructive feedback. Below, we provide a detailed point-by-point response to address the concerns raised.
>
> **W2(Narrow Trigger at Grasping)& W3 (Rollback Feasibility) & Q1 (Irreversible Actions)**
>
> **Response**: This is a crucial clarification. Our "Rollback" is a Control Rollback (***resetting the robot arm's pose***), not a Physics Rollback (reversing time or resetting the environment).
> - **Rollback is Not "Undo"**: The "rollback" in our framework is not a physical reversal of time (like "inverting gravity"). The system "performs a state rollback of k steps to retrieve a more stable, recent observation, $o_{t}$" from its memory. It then discards the intervening plan and replans from that earlier, "healthier" state.
> - **Irreversible States**: The reviewer's example of a dropped object is an irreversible failure state.  If this happens, the robot arm will continue to follow the predicted action step until the task fails, because the robotic arm can't perceive the drop action without a tactile sensor. When failed:  If the object remains within the camera's field of view and the episode has not exceeded the maximum step count, the policy will perceive the object's new position and generate a new grasp attempt based on the updated observation, and then run our "Evaluate-Imagine-Correct" loop until exceeds the maximum step count. Otherwise, if the object falls out of sight, the task is considered a failure. More detailed analysis is presented as follows.
>
> We conducted a statistical analysis of 100 failed episodes from the base policy (CogACT) across our three platforms (WidowX, Google Robot, and Real Robot) to verify why we chose to trigger at grasping time. The breakdown of failure modes is presented below:
> |Failure Stage|Approach|Grasping|Post-Grasp|
> |:-:|:-:|:-:|:-:|
> |Count (Frequency)|1|88|11|
>
> **Failures during the processing**:We thank the reviewer for raising this critical point regarding failures that occur outside the grasping moment. To address this quantitatively, we conducted a statistical analysis of 100 failed episodes from the base policy (CogACT) across our three platforms (WidowX, Google Robot, and Real Robot). The breakdown of failure modes is presented below:
> - **Approach Phase (Before Grasping): 1% (1/100) Why we don't trigger here**. As shown in the data, failures during the approach phase are remarkably rare. The SOTA VLA policies are generally robust at identifying the target and reaching its vicinity. Furthermore, defining a "wrong state" during continuous motion is $ambiguous$ compared to discrete actions. Given the low frequency of errors here, triggering interventions during approach yields diminishing returns.
> - **Grasp Initiation (The Keyframe): 88% (88/100) Validating our Trigger Design**. The vast majority of failures occur precisely at the moment of grasping (e.g., gripper misalignment, timing errors). This empirical evidence strongly validates our framework's design: by triggering the World Model intervention specifically at the keyframe, we target the dominant failure mode that accounts for 88% of task failures. This confirms that our trigger mechanism is not "too narrow," but rather highly optimized for the most critical bottleneck.
> - **Transport Phase (After Grasping)**: 11% (11/100):  *We honestly acknowledge that our current vision-based framework cannot resolve irreversible physical failures that occur post-grasp (e.g., object drop), a pure vision-based system sometimes blocks the object view and without integrating tactile or force sensors, **the agent lacks the proprioceptive feedback to realize contact has been lost***. Consequently, in such scenarios, the robot will indeed continue to execute the predicted trajectory until the task fails.  We fully agree that addressing dynamic failures during transport is the necessary next step for currently robust manipulation. ***This will require expanding our World Model to incorporate multi-modal inputs (e.g., tactile/force feedback) to perceive the "process state" beyond just visual keyframes***. We appreciate this insight, as it clearly directs our future research toward multi-modal closed-loop control.

---

> ### Author Response · Authors · 2025-11-25
> **Response to Reviewer eohM, (2/2)**
>
> **Q2 (Rollback to the Start)**:
>
> **Response**: This is a very insightful question. To answer this directly, we conducted the experiment of 96 episodes on the WidowX comparing our optimal 20-step rollback against a full "Rollback-to-Start" baseline, where the Video Model + IDM generates the complete action sequence from the first frame. The performance does not increase monotonically, as shown in table.
> |Rollback steps|15|20|25|Rollback to start|
> |-|:-:|:-:|:-:|:-:|
> |Success rate|59.4|63.5|61.5|60.4|
> We summarize this with three key reasons:
> - **Error Accumulation & Hallucination**: While short-term "imagination" is effective, video generation models suffer from "hallucinations" and temporal drift over long horizons. Generating a full episode introduces significant accumulated errors that the IDM cannot effectively resolve. The IDM's decoding accuracy degrades as the span of the generated trajectory increases, leading to imprecise actions.
> - **Open-Loop vs. Closed-Loop Control**: Relying entirely on a generated video from the start effectively turns the system into an open-loop controller. It ignores the real-time visual feedback and dynamic physical interactions of the environment. Our method retains the base VLA's closed-loop perception for the majority of execution, utilizing the World Model only for short-term, targeted corrections.
>
> Other Concern: Inference Cost: Generating a full episode via the video model is computationally expensive and prohibitively slow for online tasks. Our event-triggered approach minimizes this cost by invoking the generator only when necessary.
> Therefore, the 20-step rollback represents the "sweet spot," balancing recoverability with generation fidelity and efficiency.
>
> **W1 &W2 & W4&Q3 : Novelty & Generalization &DreamGen Comparison**
>
> **Response**: We thank the reviewer for raising this important point regarding novelty and generalization. The reviewer's perspective is appreciated, and it offers us a valuable opportunity to clarify the core contributions of our work. Unlike prior works that use World Models for continuous planning (e.g., Dreamer) or purely offline data augmentation, which typically rely on continuous, frame-by-frame rollouts to generate future states, a computationally expensive process impractical for real-time control.  To this end, we introduce a lightweight, event-triggered world model mechanism with extra (~4s) time-consuming that allows a pre-trained VLA to "self-correct" without expensive retraining, as a ***"Plug-in" and a Universal***, Online Correction framework.
> - **Generalization**: As stated in our abstract, our system intervenes "When the VLA proposes a high-stakes action... at this critical juncture". We intentionally focused on robotic grasping to validate the core "Propose-Evaluate-Imagine-Correct" loop, as ***grasping is a fundamental precursor*** to the majority of embodied manipulation tasks (e.g., folding clothes requires an initial grasp). However, it is crucial to clarify that our framework is **not inherently limited** to this domain. Theoretically, the framework is agnostic to the specific action type and can **extend to any task** involving distinct state transitions. For continuous tasks (e.g., wiping), the discrete "close gripper" trigger can be replaced by alternative state-based triggers, with the discriminator fine-tuned on the corresponding task data to ensure robust generalization. Moreover, the framework is not limited to VLAs and can be adapted to other policies, such as Diffusion Policies. This work primarily serves as a ***proof-of-concept*** for the self-correction paradigm.  Our next steps will extend this to support diverse architectures, make it more general and intelligent, with more complex, long-horizon tasks (e.g., cloth folding).
> - **Regarding DreamGen**: We sincerely thank the reviewer for the reference. This is very relevant and excellent work, and we will add the citation and discuss the distinction in the final version: DreamGen operates offline (pre-training) by using World Models to generate synthetic data. Its goal is to expand the robot's exposure to diverse scenarios, enhancing Generalization (making the robot "well-traveled"). Our framework explicitly closes the loop by coupling a discriminative gatekeeper with a generative corrector specifically for online intervention during execution failures.

---

### Official Review · Reviewer_9SzW · 2025-10-31

**Soundness:** 4
**Presentation:** 4
**Contribution:** 4
**Rating:** 6
**Confidence:** 2

**Summary:**

The paper introduces VLA-in-the-Loop, an online correction framework for VLA policies targeted at grasping. At a keyframe trigger—when the base VLA proposes gripper-close—a VLM-based discriminator judges feasibility; on predicted failure, the system rolls back k steps, invokes a video generator to “imagine” a short successful future, and feeds the imagined clip back to a unified VLA module (weight-shared with the discriminator) to decode a corrected action via an inverse-dynamics role. The approach aims to avoid continuous world-model rollouts by using an event-driven composite WM (discriminative + generative) for on-demand intervention. Experiments on SIMPLER (WidowX/Google Robot), LIBERO, and two real robots (Xiaomi, ALOHA) show consistent gains over strong baselines; ablations vary rollback depth and show robustness under online perturbations. Training uses BridgeV2-derived keyframe labels for the discriminator and successful-grasp clips for the generator;

**Strengths:**

Clear, modular intervention loop (“Propose–Evaluate–Imagine–Correct”) with an event-triggered WM that avoids continuous rollouts; well specified for grasp keyframes.

Unified discriminator/actor via shared VLM and multi-task QA formulation—neat engineering to couple evaluation and action decoding.

Solid empirical coverage across SIMPLER (tables with G/S and visual/variant suites), LIBERO, and real-world tasks; perturbation robustness tests are thoughtfully designed.

Ablation on rollback depth explains why immediate corrections can be ineffective and why earlier state restoration helps the generator produce viable plans.

**Weaknesses:**

1. VQA-style discriminator calibration. The feasibility check is framed as text-prompted VQA with labels “suitable/unsuitable”. The paper lacks calibration/ROC evidence, thresholding, and prompt sensitivity analyses: false positives trigger unnecessary rollbacks; false negatives permit failures. This is critical because the discriminator gates generation and alters control.

2. Perception–action mismatch in imagination. The generator conditions primarily on images + text; it is unclear whether proprioception and contact state are modeled or enforced. Imagined videos may depict poses infeasible for the current robot kinematics, causing a covariate gap when the actor decodes actions from pixels. A formal constraint (e.g., action-conditioned or dynamics-consistent loss) is missing.

**Questions:**

Please see weaknesses

---

> ### Author Response · Authors · 2025-11-24
> **Response to Reviewer 9SzW**
>
> We sincerely thank the reviewer for their thorough, constructive, and insightful feedback. We would like to use this opportunity to clarify the weaknesses raised in the review.
>
> **W1: VQA Discriminator Calibration**
>
> **Response**: To address the concerns regarding calibration, we conducted a comprehensive manual analysis of 200 experimental trials. We need to clear the definition of FP and FN, the discriminator only triggered when the base policy predicts the gripper should be closed. There are two situations of misclassification:
> ||Predict Consistent|Predict Inconsistent|
> |:-:|:-:|:-:|
> ||TP&nbsp;&nbsp;FP|TN&nbsp;&nbsp;FN|
> |Discriminator|144&nbsp;&nbsp;13|37&nbsp;&nbsp;6|
> |Success num|-|29&nbsp;&nbsp;2|
>
> - False Positives:  FP means discriminator agrees with base policy's action (close gripper) with a wrong prediction. In this situation (13/200=6.5%),  the robot executes the base policy without the intervention, and it represents the baseline performance floor rather than induced error.
> - False Negatives:  FN presents the discriminator incorrectly flagged a feasible state as "unsuitable" (6/200=3%). **The system triggers the rollback and generation loop, which incurs a latency penalty.As observed in our trials, the task can still succeed (e.g., 2/6 resulted in success despite the unnecessary intervention).**
>
> **Overall Accuracy**: The discriminator achieved 90.5% accuracy(TP+TN=181/200) with 3B model.
>
> **Threshold**: The VLMs frequently exhibit overconfidence, producing probability distributions heavily skewed toward the extremes (0.99 or 0.01).  There is a negligible difference between thresholds of 0.6 and 0.7, whereas the distinction between 0.99 and 0.999 is critical. Since images cannot be rendered in this response, we will visualize the ROC curve in the final version of the paper.
>
>  **Prompt Sensitivity**. Regarding prompt sensitivity and our approach differs from fragile zero-shot VQA prompting.
> - Supervised Fine-Tuning: We do not rely on the base VLM's zero-shot calibration. Instead, we fine-tuned the Qwen2.5-VL on a curated total of 102k frames. Each frame was manually or semi-automatically simple labeled as ’suitable’ or ’unsuitable’ for grasping, ensuring a balanced 1:1 ratio of positive to negative samples to prevent classification bias.
> - Prompt Stability: By training explicitly on a fixed prompt template, the model learns a robust decision boundary grounded in specific robotic affordances rather than linguistic semantics. **Our internal ablations on different prompt styles confirmed that the primary factor influencing discriminator performance is the model's spatial understanding capability, not the specific phrasing of prompt.** Generally, the spatial understanding adheres to a scaling law with model size, just as the table shows.
>
> |Prompt style|3B|7B|
> |:- | :-: | :-: |
> |Prompt base(paper)|90.1%|93.1%|
> |Prompt 2|90.3%|93.0%|
>
>     Prompt base: My task is "". Should the robot arm close its gripper in the next step?
>     Prompt 2: My task is "".  Whether you can successfully pick up the target if the gripper is closed at the current position.
>
> **W2: Perception-Action Mismatch in Imagination**
>
> **Response**: The reviewer rightfully questions if "imagined" videos respect robot kinematics. The robot arm does not have a tactile sensor, so we do not enforce an explicit dynamics loss to model the proprioception and contact state. **However, such a paradigm (video model + IDM) has been confirmed to be an effective pipeline, e.g, recent NVIDIA's work DreamGen[1].** The system ensures consistency through three mechanisms:
>
> - **Data-Driven Kinematics**: The video model is fine-tuned on 35k video clips from simulation environment and real robots, capturing the crucial moments of a successful grasp. By training on a large corpus of physically plausible videos, the generator learns to synthesize trajectories that are, by extension, visually and physically consistent with the training data.
> - **The "Inverse Dynamics" Filter**: Crucially, the generated video is not executed directly, the world model predicts where the gripper can be closed. It serves as the input to the IDM, which is trained to take the "observation sequence $O_{t^{\prime}:t^{\prime}+T}$" (the imagined video) and map it to the "action for each frame"  from the original, successful demonstrations and decodes the action sequence $a_{t':t'+T}$. The IDM,  trained on expert actions, acts as a grounding layer, it maps the visual intent of the video to the closest valid proprioceptive action, effectively filtering out minor visual artifacts.
>
> Despite extensive optimization, we acknowledge that completely eliminating kinematically infeasible poses in generated videos remains an open challenge. Residual errors can still accumulate when the IDM decodes actions from these images which direction for our future research.
>
> Jang, Joel, et al. "DreamGen: Unlocking Generalization in Robot Learning through Neural Trajectories." arXiv e-prints (2025).

---

> > ### Comment · Reviewer_9SzW · 2025-11-24
> >
> > Dear Authors,
> > Thank you for your rebuttal. Most of my concerns have been addressed. Here I update my confidence from 2 to 4, and I will carefully read other rebuttals to consider whether I need to increase my score.

---

> ### Author Response · Authors · 2025-11-24
> **We Appreciate Reviewer's Valuable Time**
>
> Dear Reviwer 9SzW:
>
>    We sincerely appreciate your quick response and the time you have dedicated to reviewing our manuscript. We hope that our revisions and explanations have adequately addressed your concerns. We are more than happy to engage in further discussion should you have any remaining questions, and we look forward to hearing from you.
>
>
> The Authors.

---

### Official Review · Reviewer_6Ngr · 2025-11-01

**Soundness:** 2
**Presentation:** 2
**Contribution:** 3
**Rating:** 4
**Confidence:** 4

**Summary:**

The paper proposes VLA-in-the-Loop, an event-triggered, online correction framework for robotic grasping on top of a base Vision-Language-Action (VLA) policy. Rather than rolling a world model (WM) continuously, the system intervenes only at high-stakes actions (specifically, when the policy proposes to close the gripper) to (i) classify whether executing the action would likely fail (via a Qwen-VL2.5-based VQA discriminator), and if risky, (ii) imagine a short successful future video using a WAN-2.1 I2V diffusion model after rolling back k frames, and (iii) decode a corrected action sequence via an inverse-dynamics module to execute the grasp. The authors claim this plug-in loop improves robustness while paying the generation cost only when needed, and report gains on several manipulation benchmarks, including LIBERO-Franka (Table 3). Key implementation details include the discriminator’s VQA formulation, a curated 102k labeled keyframe dataset with suitable/unsuitable tags, and a generator trained/fine-tuned on 33k BridgeV2 clips plus 200 real-robot videos (Xiaomi, ALOHA). Reported latency is ~0.2 s per normal step vs ~23 s for a triggered correction (≈0.95 s discriminator + 20 s video generation).

**Strengths:**

Paper provides
1. A clear systems recipe for event-triggered online correction: evaluate → (if risky) roll back → imagine → decode, targeting the exact moment where grasp outcomes hinge
2. integration of existing components (Qwen-VL-2.5, WAN-2.1, IDM) with unified QA supervision and parameter sharing, which is data/compute efficient in spirit.
3. Ablation signal that rollback depth materially affects success, aligning with intuition that one must retreat to a recoverable state before correcting.
Paper performs a number of experiments across simulation and real-robot, suggesting practical value if latency and triggering are handled carefully.

**Weaknesses:**

The paper tells an appealing “online correction” story, but the evidence is not yet strong. The trigger is too narrow (only at gripper closure), the system incurs ~23 s stalls per intervention with no statistics on how often that happens, and there are no compute-/data-fair strong baselines to isolate the value of video imagination. On the simulation benchmarks, several comparison methods are not current SOTA, which risks depressing the baselines and inflating the apparent gains.

Data / reporting consistency
* The paper does not report intervention frequency per episode, episode time distributions, or success-per-minute; only the one-off latency breakdown is given. Without these, the claim that the system is “real-time most of the time” is untested.

* Real-robot experiments lack variance / confidence intervals; online perturbation is described qualitatively, not as robustness curves (success vs. disturbance level).

Baselines and fairness
* Many simulated and real-robot baselines are not the strongest available today (e.g., more recent VLA or diffusion-action systems, stronger planners). This weakens the case that the proposed loop beats credible state-of-the-art practice.

* Missing a generator-free strong baseline under the same discriminator and a similar compute/time budget. Current gains may be due to extra capacity/data, not the “imagine-and-correct” step itself.

Method assumptions / external validity
* Triggering only at gripper closure is too restrictive; many failures start during approach / alignment. The need for rollback itself suggests the correction often arrives too late.
* The paper does not quantify physics/geometry plausibility of generated clips (contact stability, grasp quality) and does not specify fallbacks for low-confidence or time-out cases—risking confident but unsafe corrections.


Typos and other formatting errors:

1.	Line 233: “top raw” → “top row.” In the generator description: “as shown in top raw in Figure 3”.
2.	Case inconsistency in model naming + citation style. The paper uses both “WAN2.1 Wan (2025)” and “wan2.1 (Wan, 2025)”.
3.	BridgeV2 capitalization inconsistency. “Bridgev2 dataset” vs “BridgeV2” elsewhere.
4.	Line 611: Placeholder not replaced. “Figure X illustrates our 7-DoF Xiaomi Robot setup Figure 1b.”

**Questions:**

1.	Please report per-episode intervention counts (mean/median/95th), episode duration distributions, and success-per-minute vs. baselines. Do your conclusions hold under these metrics?
2.	With the same discriminator and a matched compute/time budget, how does a no-generation variant perform? This isolates the value of video imagination.
3.	What happens if you also trigger at approach / alignment, or learn a risk-based trigger? Can rollback depth be adaptive rather than fixed?
4.	Do you compute any geometry/physics consistency scores for generated clips? What is the fallback policy when generation fails or confidence is low, and how often does that occur?
5.	Can you provide success vs. disturbance magnitude/frequency curves and time-to-recover under interventions?
6.	Can you include current SOTA or widely accepted strong alternatives on at least a subset of tasks in simulation benchmarks or real-robot experiments?

---

> ### Author Response · Authors · 2025-11-24
> **Response to Reviewer  6Ngr, (1/3)**
>
> We thank the reviewer for their detailed critique. We appreciate the opportunity to clarify the misconceptions regarding latency and strengthen our evidence regarding baselines and method validity.
>
> **W1, Q1, Q5: On Latency, Efficiency, and Reporting**
>
> **Response**: We thank the reviewer for their careful attention to the latency breakdown. The ~23s intervention time mentioned in the review was indeed a significant pause, and we agree this is a critical practical concern.
>
> ||parms Disc/Gen.|latency Disc/Gen.|Avg.(Success)(WidowX/Google-VM/Google-VA/Real)|
> |-|-|-|-|
> |Base policy(CogAct)|-|-|51.8%/74.8%/61.3%/63.6%|
> |#Setting-1(Main results in Paper)|3B/5B|0.3s/4s|63.5%(+11.7)/78.5%(+3.7)/67.0%(+5.7)/76.1%(+12.5)|
> |#Setting-1(max intervention=2)|3B/5B|0.3s/4s|65.6%(+13.8)/78.7%(+3.9)/68.1%(+6.8)/78.1%(+14.5)|
> |#Setting-2(Large Model size supplement in response)|7B/14B|0.9s/20s|72.9%(+21.1)/78.9%(+4.1)/68.6%(+7.3)/81.3%(+17.7)|
>
> **Latency Optimization**: We must clarify that this ~23s figure was based on our initial experiments with a 14B parameter generator model. We have since optimized this component significantly. By utilizing a more compact 5B parameter generator (WAN 2.2), we have ***reduced the video generation time from ~20s to ~4s***, and the main results in the paper are based on this setting.  The efficiency trade-off with increased latency can be shown as #setting1 and #setting2 in the table, while larger models demonstrate strong scaling potential, *we consider Setting-1 to be highly practical and optimal balance for real-time deployment*.
>
> ||Predict Consistent|Predict Inconsistent|
> |:-:|:-:|:-:|
> ||TP&nbsp;&nbsp;FP|TN&nbsp;&nbsp;FN|
> |Discriminator|144&nbsp;&nbsp;13|37&nbsp;&nbsp;6|
> |Success num|-|29&nbsp;&nbsp;2|
>
> **Event-Triggered Efficiency**: The correction loop is event-triggered and highly selective. In our evaluation, the valid trigger rate (necessary interventions) was notably high ($TN/(TN + FN)\approx 86\\%$, 37/43),  and the average whole correction loop was activated in only 21.5% of episodes (43/200). In the remaining 78.5% of cases, the base policy executed without interruption or added latency.
>
> ||Intervention Frequency (time/Episode)|Avg.Successd(%)|Avg. Execute steps|Avg. Lantency (Second/Episode)|Success(Episode/min)|
> |:-:|:-:|:-:|:-:|:-:|:-:|
> |CogAct|-|51.1|67.8|15.7s|1.92|
> |Max=1|0.23|63.5|59.5|15.2s|2.51|
> |max=2|0.29|65.6|58.3|15.4s.|2.56|
> |max=3|0.33|65.6|60.1|16.2s|2.43|
> |max==+∞|0.55|66.7|65.7|18.1s|2.21|
>
> **Clarification on Metrics**: This is a crucial clarification. The strong quantitative results (per-episode intervention counts, episode duration, etc.) Verification of 96 episodes on the WidowX are reported in the above table. We analyzed the system performance under different maximum intervention limits (Max=1, 2, 3, +$\infty$). The results, detailed in the table above, reveal a critical insight regarding latency.
>    - **The Latency Paradox**: Why Intervention Reduces Average Time Crucially, our data shows that enabling the World Model (Max=1) actually decreases the average episode latency (15.2s) compared to the Baseline (15.7s), despite the ~4s cost of generation.
>      - **Reasoning**: The base CogACT policy operates at 0.25s per step with a maximum horizon of 120 steps. When the base policy fails (e.g., a missed grasp), it often fails to detect the error and continues executing meaningless actions until the timeout (120 steps).
>      - **Early Termination**: Our Discriminator effectively acts as ***an early-warning system***. By intervening at the critical moment, it cuts short these "doomed" trajectories. The time saved by reducing the average execution steps (from 67.8 to 59.5) effectively offsets the computational overhead of the generative model.
>   - **Success-per-Minute (Efficiency)** We shows the  "success-per-minute" matrix  to test real-world viability in above table. As shown, our method significantly outperforms the baseline. Max=2 achieves the highest efficiency at ***2.56 episodes/min*** compared to the baseline's 1.92. This confirms that the VLA-in-the-Loop framework is not just more robust, but more time-efficient in achieving successful outcomes.
>
>  **Ablation on Intervention Limits (Max=N)**.We observed that performance plateaus after Max=2:
>   - **Diminishing Returns**: Increasing from Max=2 to Max=3 yields no improvement in success rate (65.6%) but increases latency (16.2s) and lowers efficiency. This indicates that remaining "hard cases" cannot be solved simply by adding more one retry.
>   - **Unbounded Intervention**: Setting Max=∞ provides only a marginal success gain (66.7%). While allowing unlimited interventions helps resolve a small number of residual hard cases, it causes a spike in trigger frequency (0.55) and increases average latency to 18.1s. This significant efficiency penalty renders the unbounded setting less practical for real-world deployment compared to the bounded configurations.

---

> ### Author Response · Authors · 2025-11-24
> **Response to Reviewer 6Ngr, (2/3)**
>
> **W4&Q2: Isolating the Value of Imagination**
>
> ||Avg.|Spoon on Towel|Carrot on Plate|Stack Cube|Put Eggplant in Basket|
> |-|:-:|:-:|:-:|:-:|:-:|
> |CogACT|51.8|71.7|50.8|15.0|67.5|
> |CogACT + Noise|52.1|70.8|50|16.7|70.8|
> |CogACT + Ours|63.5|75.0|58.3|37.5|83.3|
>
> **Response**: To address the concern that gains might come from "extra capacity" and isolate the specific contribution of the generated video, we conducted a controlled experiment. ***Since our base policy is deterministic, utilizing the Discriminator solely to trigger a rollback results in a deadlock***: the robot reverts to the previous state ($o_{t-k}$), receives the exact same observation, and repeats the identical failing action sequence. Therefore, to construct a functional "No-Generation" baseline, ***we introduced action noise to break this loop***. Instead of guiding the robot with an imagined video after rollback, we injected Gaussian noise into the base policy’s predicted actions to force a trajectory deviation.
> Results (Value of Imagination): As shown in the table above, simply perturbing the policy ("Noise Variant") offers no benefit, whereas our Generative approach **yields significant gains**.
>
> **Conclusion**: The "No-Generation" variant (52.1%) fails to improve upon the baseline (51.8%), confirming that rollback alone is insufficient. The significant performance leap to 63.5% is directly attributable to the World Model's imagined video, which provides the specific, dense visual guidance necessary to correct the trajectory.
>
> **W5, Q3 : On Trigger Mechanism, Generalize and Adaptive Rollback Depth**
>
> **Trigger Mechanism and Generalize**: This is a valid point. As stated in our abstract, our system intervenes "When the VLA proposes a high-stakes action... at this critical juncture". Our work is intentionally focused on "robotic grasping tasks" to validate the core "Propose-Evaluate-Imagine-Correct" loop, as ***grasping is a fundamental precursor*** to the majority of embodied manipulation tasks (e.g., folding clothes requires an initial grasp). However, it is crucial to clarify that our framework is ***not inherently limited to this domain***. Theoretically, the framework is agnostic to the specific action type and can extend to any task involving distinct state transitions.  As you note, we can choose multiple trigger timings, e.g, approach and alignment, but such a multiple candidate trigger design will ***decrease the intervention trigger precision***. In the current framework, we prefer to choose a single one obvious state change and high-stakes action.
>
> |Failure Stage|Approach|Grasping|Post-Grasp|
> |:-:|:-:|:-:|:-:|
> |Count (Frequency)|1|88|11|
>
> **Failures during the processing**: We thank the reviewer for raising this critical point regarding failures that occur outside the grasping moment. To address this quantitatively, we conducted a statistical analysis of 100 failed episodes from the base policy (CogACT) across our three platforms (WidowX, Google Robot, and Real Robot). The breakdown of failure modes is presented below:
> - **Approach Phase (Before Grasping): 1% (1/100) Why we don't trigger here**. As shown in the data, failures during the approach phase are remarkably rare. The SOTA VLA policies are generally robust at identifying the target and reaching its vicinity. Furthermore, defining a "wrong state" during continuous motion is $ambiguous$ compared to discrete actions. Given the low frequency of errors here, triggering interventions during approach yields diminishing returns.
> - **Grasp Initiation (The Keyframe): 88% (88/100) Validating our Trigger Design**. The vast majority of failures occur precisely at the moment of grasping (e.g., gripper misalignment, timing errors). This empirical evidence strongly validates our framework's design: by triggering the World Model intervention specifically at the keyframe, we target the dominant failure mode that accounts for 88% of task failures. This confirms that our trigger mechanism is not "too narrow," but rather highly optimized for the most critical bottleneck.
> - **Transport Phase (After Grasping)**: 11% (11/100):  *We honestly acknowledge that our current vision-based framework cannot resolve irreversible physical failures that occur post-grasp (e.g., object drop), a pure vision-based system sometimes blocks the object view and without integrating tactile or force sensors, **the agent lacks the proprioceptive feedback to realize contact has been lost***. Consequently, in such scenarios, the robot will indeed continue to execute the predicted trajectory until the task fails.  We fully agree that addressing dynamic failures during transport is the necessary next step for currently robust manipulation. ***This will require expanding our World Model to incorporate multi-modal inputs (e.g., tactile/force feedback) to perceive the "process state" beyond just visual keyframes***. We appreciate this insight, as it clearly directs our future research toward multi-modal closed-loop control.

---

> ### Author Response · Authors · 2025-11-24
> **Response to Reviewer 6Ngr, (3/3)**
>
> **Continue: W5, Q3 : Adaptive Rollback Depth**
>
> **Adaptive Rollback Depth**: We agree with the reviewer that adaptive rollback is a promising direction. However, we did not adopt this strategy in the current version because ***defining the "Ground Truth" of which historical frame provides the optimal observation for the World Model is non-trivial, it is analytically difficult to determine a priori which specific past frame maximizes the probability of successful recovery without extensive supervision***. We sincerely appreciate the reviewer pointing this out and view adaptive rollback as a valuable next step for the community. ***We would like to thank the reviewer for any specific suggestions might have regarding self-supervised signals for determining optimal depth***. And we believe this does not diminish the core contribution of our framework is a lightweight, event-triggered mechanism with extra (~4s) time-consuming that allows a pre-trained VLA to "self-correct" without expensive retraining as a "Plug-in" and a Universal, Online Correction framework.
>
> **W6&Q4 : WM Physics Consistency and Fallback**
>
> |              | PSNR | SSIM | FVD |
> | -------------- | ----- | --------------- | --------------- |
> | Cosmos-Predict2.5/action-cond   | 24.95 | 0.85            | 146           |
> | Wan-2.2(5B) finetune | 21.56 | 0.81            | 163           |
> | Wan-2.1(14B) finetune | 26.27 | 0.90            | 131          |
>
> **Response**: To rigorously evaluate the quality and physical plausibility of our generated videos, we conducted a quantitative comparison against ground truth using the official BridgeV2 test set. We ***randomly sampled 50 episodes*** and compared the generations from our fine-tuned model against Cosmos-Predict2.5-2B, a SOTA action-conditioned video generation baseline. As shown in the table above, ***our model achieves performance on par with Cosmos, notably achieving this fidelity even without explicit action conditioning***. *Since prior work such as DreamGen has already verified the feasibility of using Cosmos-generated synthetic data for effective policy training*, our model’s comparable performance strongly validates that it meets the geometric and physical requirements necessary for reliable policy correction.
>
> **Fallback Policy**: If the video generation fails (e.g., blur) or if the IDM fails to decode a confident action sequence from the generated clip (indicating low quality) . We implement an ***iterative Retry Mechanism***. Specifically, if the generated plan is deemed invalid or if the Discriminator continues to flag the state as "Unsuitable" after the initial correction attempt, the system automatically re-triggers the "Propose-Evaluate-Imagine-Correct" loop. This process repeats up to a pre-set maximum intervention limit (e.g., 2 interventions), allowing the system to leverage the stochastic nature of the generative model to re-sample a potentially higher-fidelity trajectory and action sequence.
>
> Furthermore, we statistically analyze the frequency of generation failure as follows:
>
> ***As noted, the correction loop was triggered in 21.5% of episodes (43 out of 200 trials).***
>
> To determine how often the generator itself fails (e.g., hallucinations or physics violations), we manually inspected the generated videos for all intervention cases. ***Out of 43 interventions, 12 resulted in final task failure.***
>
> **Cause Analysis**: Among these failures using the WAN2.2 5B model, only 4/43 cases, while the 14B model decreased to 2/43, tatally (4/200 $\to$ 2/200), were caused by severe structural or positional deviations in the generated video.
>
> **Q6: Compared with SOTA methods**
>
> |              | Avg. Success | Spoon on Towel | Carrot on Plate | Stack Cube | Put Eggplant in Basket |
> |:-: | :-: | :-: | :-: | :-: | :-: |
> | X-VLA(LoRA)*    | 64.6% | 83.3           | 75.0            | 20.8       | 79.2                   |
> | X-VLA(LoRA)+ours | 75.0% | 83.3         | 79.2            | 54.1       | 83.3               |
>
> **Response**: We thank the reviewer for the suggestion. We would like to clarify that VLA-in-the-Loop is designed as a ***Universal, Model-agnostic Plug-in rather than a standalone policy that competes directly with base models***.
> Limited by response time, we apply our framework to one currently well-known open-source VLA model (X-VLA  with LoRA Adapters) , which can bring about stable performance improvement in the above table. We will apply more VLA baselines and compare the results which can be updated to the final version.

---

### Official Review · Reviewer_18c6 · 2025-11-04

**Soundness:** 3
**Presentation:** 3
**Contribution:** 2
**Rating:** 6
**Confidence:** 3

**Summary:**

This paper introduces VLA-in-the-Loop, a novel framework that integrates Vision-Language-Action (VLA) models with a composite World Model (WM) to enable real-time policy correction for robotic manipulation.

Traditional VLA systems (e.g., RT-2, CogACT) map visual and textual inputs to robotic actions through imitation learning but lack mechanisms for online correction once errors occur. Meanwhile, World Models have strong predictive abilities but are computationally expensive due to continuous rollouts.

The proposed method bridges these paradigms by introducing an event-triggered, lightweight correction loop:
1) When a high-stakes action (e.g., closing a gripper) is proposed,
2) A discriminative module (a fine-tuned Vision-Language Model like Qwen-VL 2.5) evaluates whether the action is feasible.
3) If failure is predicted, a generative module (a video diffusion model, e.g., WAN2.1) “imagines” a short video showing a successful future trajectory.
4) This imagined trajectory is then fed back to the VLA, guiding it to produce a corrected and more robust action.

This “Propose–Evaluate–Imagine–Correct” loop provides a form of online intervention that corrects policy execution in real time without requiring full-sequence simulation.

**Strengths:**

S1) Instead of using world models for continuous prediction, the paper redefines them as on-demand correctors, activated only at critical decision points. This is a clever shift from passive supervision to active, event-driven guidance.
S2) The separation into a discriminator (judge) and a generator (imaginer) makes the system modular and interpretable, with clear functionality and training objectives for each part.
S3) Addresses real-world limitations of robotic manipulation pipelines — namely, lack of online correction and high cost of continuous predictive reasoning.

**Weaknesses:**

W1) The provided description does not clarify how often the correction loop is triggered, its computational overhead, or quantitative improvements across benchmarks. The efficiency vs. accuracy trade-off needs clearer measurement.
W2) The framework assumes that the system can reliably identify “critical moments” (e.g., grasp initiation). Errors in key-frame detection could undermine correction timing.
W3) Most examples focus on grasp correction. It remains uncertain how well the system generalizes to other manipulation types (e.g., pushing, insertion, tool use).
W4) The discriminative model’s success heavily depends on how well Qwen-VL generalizes to unseen grasp scenes. The approach may struggle in low-text or low-visibility scenarios without explicit grounding.

**Questions:**

How often is the world model triggered in typical tasks, and what is the average latency introduced by the “imagine–correct” loop compared to baseline inference?

What is the false positive/negative rate of the discriminative module in predicting failure? How sensitive is performance to misclassification?

Can the framework extend to tasks that lack clear discrete keyframes (e.g., continuous tool manipulation, deformable-object handling)?

How does the system perform as the size or complexity of the world model increases? Would a single joint model (rather than modular discriminative + generative components) improve stability?

How does this approach compare with recent reflection-based or self-correcting robot architectures (e.g., Phoenix 2025, Reflexion, or LVLM-based reasoning controllers)?

---

> ### Author Response · Authors · 2025-11-24
> **Response to Reviewer 18c6, (1/2)**
>
> We sincerely thank the reviewer for their thorough, constructive, and insightful feedback and for recognizing our event-driven, modular approach as "clever" and "interpretable." We address the specific concerns of  the weaknesses (W) and answer the questions (Q) below.
>
> **W1 & Q1, Q4: Frequency, Latency, Model Size and  Computational Overhead**
>
> **Response**: The activation frequency of our correction loop is dynamic and depends on the performance of the base policy, the better the policy (i.e., a better policy triggers fewer interventions); To quantify this, we analyzed 200 trials across our multi-task benchmarks (WidowX, Google Robot, and Real Robot) of our event-triggered loop on base policy(CogAct):
>
> || Predict Consistent| Predict Inconsistent |
> |:-: | :-: | :-: |
> | | TP&nbsp;&nbsp;&nbsp;&nbsp;&nbsp;&nbsp;&nbsp;&nbsp;FP|TN&nbsp;&nbsp;&nbsp;&nbsp;&nbsp;&nbsp;&nbsp;&nbsp;FN|
> | Discriminator  | 144&nbsp;&nbsp;&nbsp;&nbsp;&nbsp;&nbsp;&nbsp;13|37&nbsp;&nbsp;&nbsp;&nbsp;&nbsp;&nbsp;&nbsp;&nbsp;6 |
> | Success num |  -&nbsp;&nbsp;&nbsp;&nbsp;&nbsp;&nbsp;&nbsp;&nbsp;- |29&nbsp;&nbsp;&nbsp;&nbsp;&nbsp;&nbsp;&nbsp;&nbsp;2 |
>
> **WM Trigger Frequency**: The correction loop is event-triggered and highly selective. In our evaluation, the valid trigger rate (necessary interventions) was notably high (***TN/(TN + FN)=86%***, 37/43),  and the average whole correction loop was activated in only 21.5% of episodes (43/200). In the remaining 78.5% of cases, the base policy executed without interruption or added latency.
>
> ||Parms Disc/Gen.|Latency Disc/Gen.|Avg.(Success)(WidowX/Google-VM/Google-VA/Real)|
> |-|-|-|-|
> |Base policy(CogAct)|-|-|51.8%/74.8%/61.3%/63.6%|
> |#Setting-1(Main results in Paper)|3B/5B|0.3s/4s|63.5%(+11.7)/78.5%(+3.7)/67.0%(+5.7)/76.1%(+12.5)|
> |#Setting-1(max intervention=2)|3B/5B|0.3s/4s|65.6%(+13.8)/78.7%(+3.9)/68.1%(+6.8)/78.1%(+14.5)|
> |#Setting-2(Large Model size supplement in response)|7B/14B|0.9s/20s|72.9%(+21.1)/78.9%(+4.1)/68.6%(+7.3)/81.3%(+17.7)|
>
> *The Disc/Gen denotes the module of WM Discriminator and WM Generator.*
>
> **Model size & Latency&Trade-off**: We performed a comprehensive ablation study across different model configurations (Setting-1, Setting-2) to identify the optimal operating point.
>   - For #setting1, we utilize a 3B Discriminator (0.3s) and a 5B Generator (4s), achieving a +11.7% success rate improvement on WidowX (63.5% vs. 51.8% baseline) while keeping the intervention cost at ~5 seconds with incurred in 21.5% of episodes, we consider this the optimal balance for real-time deployment.
>   - For #setting2, scaling to 7B/14B models boosts success by +21.1% but increases latency to ~21s. While this setting demonstrates the strong scaling potential of our VLA-in-the-Loop framework (i.e., better foundation models yield better policies), but less practical for real-time use. This justifies our choice of Setting-1 for the main paper.
>
> ***Conclusion***: While larger models demonstrate strong scaling potential, we consider Setting-1 the optimal balance for real-time deployment.
>
> **W2, W4 & Q2: Discriminator Reliability**
>
> **Response**: We recognize that the system relies heavily on the Discriminator's ability to correctly identify "critical moments." To validate this, we evaluated the 200 trials mentioned above:
>
> **Overall Accuracy**: The discriminator correctly classified the feasibility of the grasp in (TP+TN=144+37=181) out of 200 cases (***90.5%***).
> In the failed prediction (19), 13 cases are consistent with the base policy, while 6 cases are not.
> There are two situations of misclassification:
> - **Sensitivity to False Positives**:  *FP means the discriminator agrees with the base policy's action (close grigger) with a wrong prediction*. In this situation (13/200=6.5%),  the robot executes the base policy's raw action without the world model, and it represents the baseline performance floor rather than an induced error.
> - **Sensitivity to False Negatives**:  *FN presents the discriminator incorrectly flagged a feasible state as "unsuitable"* (6/200=3%). For this, the system triggers the rollback and generation loop, which incurs a latency penalty, but as observed in our trials, the task can still succeed (e.g., 2/6 resulted in success despite the unnecessary intervention).
>
> To further mitigate these edge cases, our Retry Mechanism (max intervention=2) provides a safety net, allowing the system to recover from initial prediction errors and boosting overall robustness.

---

> > ### Author Response · Authors · 2025-11-24
> > **Response to Reviewer 18c6, (2/2)**
> >
> > **W3Q3:  Generalization Beyond Grasping**
> >
> > **Response**: This is a valid point. As stated in our abstract, our system intervenes "When the VLA proposes a high-stakes action... at this critical juncture". Our work is intentionally focused on "robotic grasping tasks" to validate the core "Propose-Evaluate-Imagine-Correct" loop, as ***grasping is a fundamental precursor*** to the majority of embodied manipulation tasks (e.g., folding clothes requires an initial grasp). However, it is crucial to clarify that our framework is *not inherently limited to this domain*. Theoretically, the framework is agnostic to the specific action type and *can extend to any task involving distinct state transitions*. For continuous tasks (e.g., wiping), the discrete "close gripper" trigger can be replaced by alternative state-based triggers, with the discriminator fine-tuned on the corresponding task data to ensure robust generalization. Generalizing this framework to other manipulations like pushing or folding clothes is a very exciting and logical next step, as noted in our future work.
> >
> > **Q4：Model Joint vs. Modular**
> >
> > **Response**: We decoupled the framework into two modules to establish an ***event-triggered, on-demand system***， which employs a discriminator (~0.3s) to determine whether it is necessary to invoke the more resource-intensive world model generator. Conversely, a joint design—similar to existing world model approaches would necessitate *executing the generation step at every instance, resulting in excessive resource consumption*. For experiments on the size of world models can refer to W1 & Q1, Q4.
> >
> > **Q5: Comparison with Reflection-Based/Self-Correcting Architectures:**
> >
> > **Response**: We thank the reviewer for highlighting these relevant works. While we share the goal of self-correction, our approach differs fundamentally in design philosophy and universality.
> >   - **Universal Plug-in Framework vs. Specialized Architectures**: Unlike Phoenix 2025, which relies on a customized motion-based self-reflection pipeline, or LVLM-based controllers (e.g., RoboReflect) that require dedicated modules to parse high-level semantic text into executable control signals, our "on-demand" approach is more efficient for real-time deployment, which is designed as a universal, plug-in framework. Although we validated it on VLA models in this paper, the architecture is theoretically compatible with any robotic policy. It functions as an external "corrector" that does not require structural adaptation or retraining of the base policy.
> >   - ***Future Work***: Due to the specialized nature of these baselines compared to our agnostic plug-in design, a direct experimental comparison was not feasible within the rebuttal timeframe. However, our immediate next steps are to demonstrate this universality: we plan to extend the framework to support arbitrary policy architectures (beyond just VLAs) and diverse manipulation tasks (beyond grasping) to further validate the generalization and superiority of our plug-in design.
> >
> >   **W4-VLM Reliance**
> >
> > **Response**: The performance of the Qwen-VL 2.5 is indeed critical. However, we do not rely on the base VLM's zero-shot calibration. Instead, we fine-tuned the Qwen2.5-VL model on a curated total of 102k frames, with 100k from a BridgeV2 subset and 2k from real-world trials (Xiaomi Robot and ALOHA). Each frame was manually or semi-automatically simple labeled as ’suitable’ or ’unsuitable’ for grasping, ensuring a balanced 1:1 ratio of positive to negative samples to prevent classification bias.

---

> > > ### Comment · Reviewer_18c6 · 2025-11-24
> > >
> > > Thank you for the explanation. I will retain my original score, but I’ve increased my confidence. Best of luck.

---

> ### Author Response · Authors · 2025-11-24
>
> Dear Reviewer 18c6：
>
>   Thank you for your kind feedback and for taking the time to further consider our revisions. We sincerely appreciate your increased confidence in our work, and we are glad that our responses were helpful in clarifying the points you raised.
>
> While we understand that you have decided to retain your original score, we appreciate your encouragement and hope that the manuscript in its current form contributes meaningfully to the field, and we remain open and happy to consider any further suggestions you might have.
>
> Thank you again for your constructive input and for supporting the improvement of our paper.
>
> Best regards.
>
> The Authors.

---

### Author Response · Authors · 2025-11-24
**General Response to Reveiwers**

Dear Reviewers:

We sincerely thank the reviewers for their time and constructive comments. These insights have significantly helped us clarify our contributions and improve the quality of our manuscript. In this revision, we have addressed all concerns and conducted extensive additional experiments. A summary of our major updates is provided below:

- **Quantitative Efficiency & Scaling Analysis**: We conducted extensive statistical experiments (N=200 trials) to quantify the system's operational metrics. We have added data on the intervention frequency (triggered in \~21.5% of episodes), latency analysis (\~4s for generation vs. 0.3s for discrimination), and an efficiency vs. accuracy trade-off study. We also provide granular statistics, including per-episode intervention counts, episode duration distributions, and success-per-minute metrics, confirming the system maintains real-time performance for the majority of operation. Furthermore, we validated the Scaling Law of our framework, showing that larger backbone models (7B/14B) yield significantly higher success rates.

- **Discriminator & Generator Validation**: We rigorously validated the effectiveness and sensitivity of the Discriminator. Our confusion matrix analysis confirms high classification accuracy (90.5%) and a low false-positive rate (6.5%), ensuring the system acts as a reliable gatekeeper. We also analyzed the physical plausibility of the Generator, manually inspecting failure cases to confirm that the imagined videos are effective for robotic control.

- **SOTA Comparisons & Framework Positioning**: We extended our evaluation to include current SOTA open-source VLA models (e.g., X-VLA) to demonstrate our method's ability to boost strong baselines. We also clarified the distinction between our event-triggered, visual plug-in and existing reflection-based or self-correcting architectures.

- **Generalization Beyond Grasping**: We clarified that while our method was validated on grasping—the most fundamental robotic action—the framework is not limited to gripper state changes. It is designed to generalize to any task with distinct state transitions (e.g., contact initiation in pushing, alignment in tool use).

**Limitations & Future Directions**: We honestly acknowledge the limitations identified by the reviewers, which serve as clear directions for our future work:

- **Process Failure Perception**: Our current vision-based system cannot robustly detect irreversible failures during transport (e.g., object dropping). This highlights the need for integrating multi-modal sensing (tactile/force) in the next iteration.

- **Adaptive Rollback depth**: We currently use a fixed rollback step. We agree that an adaptive rollback strategy, tailored to different robot kinematics and policy dynamics, would be more optimal. While the definition of Ground Truth of target depth should be carefully designed.

- **Universal Plug-in Vision**: While validated on VLA architectures, our method is designed as ***a universal, on-demand Plug-in Self-Correction*** Framework. Our next steps will extend this to support diverse architectures (such as Diffusion Policies) and cover more complex, long-horizon tasks (e.g., cloth folding).

We believe these additional experiments and clarifications address the core concerns raised, demonstrating that VLA-in-the-Loop is a robust, efficient, and scalable solution for online robotic policy correction.

---

### Meta-Review · Area_Chair_826k · 2026-01-07

**Summary:**

This paper proposes an event-triggered world-model plug-in (“Propose–Evaluate–Imagine–Correct”) that detects high-risk grasp moments and, when necessary, invokes short-horizon video imagination coupled with inverse dynamics to guide correction. Reviewers initially raised concerns regarding real-time practicality, intervention frequency and latency statistics, baseline fairness and isolation of the imagination component, generalization beyond grasping, and the physical consistency of generated trajectories. While the rebuttal addresses several of these issues with additional analysis and experiments, key concerns remain particularly in the restricted application setting in which multiple modules are task-specific and kind of "hardcoded". More broadly, given that world models are generally expected to support flexible, task-agnostic reasoning and planning, the current formulation appears narrowly specialized.

**Reviewer Concerns:**

Concerns largely addressed by the rebuttal
- [18c6] Efficiency, trigger frequency, and latency stats; Discriminator reliability and sensitivity; VLM dependence
- [6Ngr] Efficiency, trigger frequency, and latency stats; clarification and reporting details; expanded baselines
- [9SzW] more metrics; perception–action mismatch in imagination
- [eohM] Rollback feasibility & rollback to start; Generalization beyond grasping; comparison to DreamGen

Concerns still outstanding / partially answered;
- [18c6] Generality beyond grasping; Dependence on accurate event detection
- [6Ngr] Breadth of baselines across all tasks; robustness curve
- [9SzW] kinematic infeasibility as open challenges
- [eohM] Novelty justification

**Reviewer Scores:**

- [18c6] likely remain at 6
- [6Ngr] likely remain at 4
- [9SzW] likely remain at 6
- [eohM] likely remain at 4

---

### Decision · Program_Chairs · 2026-01-26

Reject